# Loss-of-function variants in the CAPN1 activator *CD99L2* cause X-linked spastic ataxia

Most patients with a rare movement disorder (MD) do not receive a molecular diagnosis, and the underlying genetic variants and mediating genes remain elusive. Here, we evaluate the diagnostic accuracy of conventional and next-generation sequencing-based genetic testing strategies in a cohort of 2,811 individuals with ataxia, spastic paraplegia and dystonia. Exome sequencing establishes genetic diagnoses in 19.3% of cases, and specificity of phenotypic features and age at testing are positive predictors. Genome analysis 'beyond the exome' increases the diagnostic yield by 7.5%, mostly due to the improved detection of structural variants and repeat expansions. Unsolved cases are included in the Solve-RD cohort and subjected to gene-burden analysis, providing evidence for loss-of-function variants in X-chromosomal *CD99L2* causing spastic ataxia. Cellular studies show that the transmembrane protein CD99L2 occurs mainly in a ubiquitinated form and serves as an activating interactor of the calcium-dependent protease CAPN1. Ablation of cytoplasmic or extracellular domains of CD99L2 leads to its intracellular mislocalization and abrogation of its interplay with CAPN1. Transcriptome analysis in CD99L2 patient-derived fibroblasts reveals synaptic function-specific disturbances. Impaired CAPN1 activation and dysregulation of downstream neuronal pathways constitute the likely molecular cause for neurodegeneration.

About 70% of the ~10,000 unique rare diseases (RDs) are believed to have a genetic cause[1]. However, the genetic aetiologies of more than half of these diseases remain unknown[1]. Depending on the disease entity and testing strategy applied, the diagnostic yield varies between 20 and 70%[2,3]. Strategies to increase diagnostic rates for patients include streamlined genome sequencing (GS) and phenotyping, which also enables the discovery of novel gene-disease associations by cohort-based statistical approaches[3–5]. In Germany, a structured diagnostic process that involved multidisciplinary expertise has been established at centres for rare diseases, and was evaluated at the national level through the TRANSLATE-NAMSE study[6,7]. At the pan-European level, genomic data collection of individuals affected by rare diseases has been substantially advanced by the Solve-RD initiative, through European Reference Networks[8,9]. Associated infrastructures

such as RD-Connect GPAP[10], can accelerate rare disease gene discovery by matchmaking of a prioritised candidate gene in relevant disease cohorts, e.g. Rare Neurological Diseases (RND, https://www.ern-rnd.eu/). The latter include rare movement disorders (MDs), that are a prototypical example of a clinically and genetically heterogenous group of disorders with a high rate of negative testing results (38–88%)[11–13], which could be partially explained by an enrichment of sequence variations that are difficult to detect with standard next-generation sequencing analysis, such as tandem repeat expansions[14,15]. Systematic evaluation of results from conventional fragment analysis (FA) for repeat expansion testing, exome sequencing (ES), genome sequencing (GS), and combinations thereof, can guide genetic testing in routine diagnostics. Combined with standardised clinical information, these data enable the systematic evaluation of factors impacting

✉ e-mail: jonaszjeremiasz.weber@rub.de; tobias.haack@med.uni-tuebingen.de

the diagnostic yield towards improved expectation management of patients and referring clinicians.

A better mechanistic understanding of rare disease biology by in-depth characterisation of novel disease-associated factors is key for the development of targeted therapy and bears the potential to modulate pathways for the treatment of common disease entities. Calpain biology serves as an excellent example, with components and pathology-related activation disturbances being linked to ultra-rare disorders but also Alzheimer's and Parkinson's disease, with the full set of mediating factors and cellular interactions awaiting full definition[16–18]. For instance, previous affinity capture-based mass spectrometry found the transmembrane protein CD99 antigen-like 2 (CD99L2) as an interactor of calpain-1 (CAPN1), a calcium-dependent cysteine protease involved in neuronal plasticity, and linked to forms of recessive spastic paraplegia and cerebellar ataxia[19,20]. Similar to CD99, CD99L2 plays a role in leucocyte transmigration through endothelial cell-cell contacts[21]. However, besides an association of the SNP rs11796490 with autism spectrum disorder[22], and a single case observation in cerebral palsy[23], it has so far not been linked to neurological disease.

In this work, we systematically assess the diagnostic yields of conventional and next-generation sequencing-based testing strategies across a large cohort of individuals with ataxia, spastic paraplegia, and dystonia, and identify factors predictive of establishing a firm molecular diagnosis. We perform a gene-burden analysis on in-house sequencing datasets from a phenotypically diverse collection of 13,132 individuals and subsequent matchmaking to provide evidence that loss-of-function variants in *CD99L2* cause X-linked spastic ataxia. We use overexpression cell models and patient-derived fibroblasts to characterise the function of CD99L2 and its interplay with the calpain system and to decipher the pathomolecular consequences of CD99L2 dysfunction. Our investigations reveal a reciprocal interaction between CD99L2 and CAPN1, which involves ubiquitination as a regulatory posttranslational modification (PTM). Loss of CD99L2 in patients' fibroblasts triggers transcriptional dysregulation of genes linked to neuronal and synaptic function. These disturbances likely represent the molecular cause of neurodegeneration triggered by the absence of CD99L2 in patients.

## Results

### Definition of cohort, diagnostic yield, and predictive factors

Within 6 years, 2811 individuals [of these, 2337 adults (83.1%) and 474 children (16.9%)] with a suspected rare movement disorder were referred for genetic testing from 26 sites across Germany. A total of 7241 genetic tests were performed, including 4714 FAs in 1067 cases, 2041 ES, and 486 GS (Supplementary Fig. 1a). The testing strategy initially consisted of a staged approach of FAs followed by ES, but then evolved into a prospectively evaluated first-line GS approach with subsequent FA to exclude or confirm specific repeat expansions. About 12% of cases had received previous gene panel-based testing with inconclusive findings.

The majority of cases studied by ES and/or GS were investigated under a multidisciplinary diagnostic care concept based on clinical expertise at university hospital centres for rare diseases[24]. Clinical information were collected using standardised human phenotype ontology (HPO) terms, and based on key clinical features, affected individuals were assigned to the disease categories ataxia, spasticity, dystonia and combinations thereof (for details see Online methods). The positioning of individuals according to their original HPO annotations was visualised by projection into a clinical feature space. Most patients within the same disease group were closely located to one another. However, there was some overlap among clusters (Fig. 1a), consistent with the clinical observation that categorising patients into major MD disease groups may oversimplify their conditions.

### Diagnostic yield of fragment analyses and exome sequencing ranges from 10.9 to 19.3%

A total of 4714 FAs were performed on 1067 subjects, most of them with ataxia (n = 988; 92.6%), resulting in a diagnosis in 116 cases (10.9%). The most frequently observed expansions were *FXN* (25%; Friedreich ataxia, MIM #229300), *ATXN3* (17.2%; SCA03, MIM #109150), and *CACNA1A* (16.4%; SCA06, MIM #601011). The main entities investigated by ES were ataxia (n = 899; 44%), spasticity (n = 573; 28%), and dystonia (n = 265; 13%), as well as individuals with mixed MDs (n = 304; 15%). Firm diagnoses (likely pathogenic or pathogenic variants) were established in 394 (19.3%) cases, with the highest (28.1%) diagnostic yield in the spasticity group and the lowest (12.1%) in dystonia patients (Supplementary Fig. 2c). The gene-specific modes of inheritance observed in solved cases were dominant (51%), recessive (42%), X-linked (4%), and mitochondrial (3%), and the most frequently affected genes were *SPAST* (n = 46; 16%), *SPG7* (n = 19; 6.6%) and *CACNA1A* (n = 15; 5.2%). Further details of the cohort and genetic results for ES are shown in Supplementary Figs. 2 and 3.

### Genome sequencing increases diagnostic yield by ~7.5%

Using a bioinformatic pipeline optimised for the detection of complex variation [repeat expansions (RE), structural variants (SV), copy-number variations (CNV)] and interpretation 'beyond the exome'[25], the added diagnostic yield and clinical feasibility of diagnostic genome sequencing were assessed in a total of 486 MD individuals [ataxia (n = 219), dystonia (n = 19), spasticity (n = 134), and combined MDs (n = 114)]. This included patients in whom a diagnosis could not be established after previous FA and/or ES (n = 272; 56%), as well as n = 214 (44%) MD individuals who were investigated in a GS first approach. In the latter, the diagnostic yield was higher than in the ES first cohort (25.2% versus 21.96%, respectively; Fig. 1b). In 200 patients with GS and previous ES plus optional FA, the diagnostic yield was substantially lower but still remarkable with 10.5% (21 solved cases, Fig. 1b). Of the reported clinically relevant findings in these cases, 15 (71%) had been missed in previous analyses, and an additional one (5%) has been technically detected but did not meet the criteria of being formally classified as a likely pathogenic or pathogenic variant at the time of initial testing. The majority of missed variation included SVs and REs that could only be detected by GS and which were accordingly enriched in the second-line GS cohort [16.5% (11/67) GS-first vs 33.3% (9/27) second-line GS; P = 0.09475, Fig. 1c]. However, the added diagnostic sensitivity of GS of about 7.5% encompassed all types of genomic variation, including small variants. Illustrative examples can be found in the Supplementary information, together with a detailed description of the GS cohort in Supplementary Figs. 1 and 4.

### Specificity of HPO terms, disease group and age at diagnosis are positive predictors for establishing a molecular diagnosis

We next aimed to assess whether factors beyond the genetic tests used could influence the diagnostic rate. A multivariate regression analysis was performed to investigate the impact of various factors, including specific subgroups of MDs, age at genetic testing, and the number of HPO terms, as well as the mean specificity of all assigned human phenotype ontology (HPO) terms per patient (derived from the relative position of HPO terms in their HPO branch). Least absolute shrinkage and selection operator (LASSO) analysis indicated a high mean specificity of HPO terms, young age at genetic testing, as well as a combined MD including spasticity as predictive parameters for establishing a firm genetic diagnosis (Fig. 1d, e). While our model was trained on the exome cohort [AUC ROC on test data 0.7 (95% CI: 0.64–0.761)], it was able to discriminate between solved and unsolved cases in the genome cohort [AUC ROC 0.609 (95% CI: 0.532–0.687), Supplementary Fig. 5].

### Cohort analyses discover novel disease gene

**Gene-burden analysis infers three known and one novel genetic association.** Taking advantage of available standardised expert-

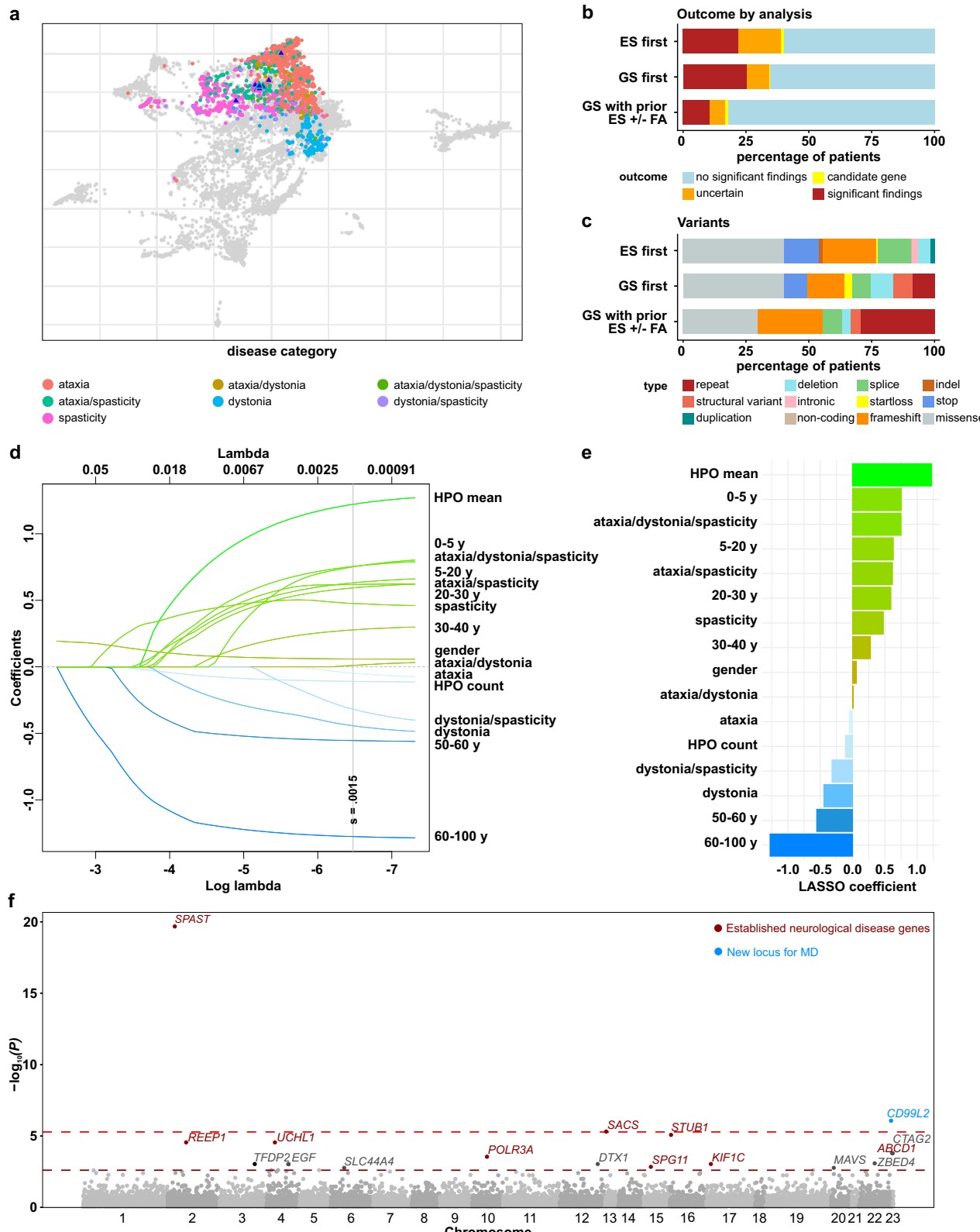

curated phenotype and sequencing datasets, we next performed a gene-burden analysis to identify genetic associations between coding genes and MD phenotypes. To account for clinical heterogeneity among cases sharing the same genetic aetiology as observed with established MD entities (Fig. 1a), we merged the different MD subgroups into one with the aim of boosting power to identify statistical genetic associations. The curated dataset comprised 2287 MD cases

and 10,845 controls. Using the two-sided Fisher exact test, gene-wise association analyses were conducted for high-quality protein-truncating variants (PTVs), and a combined set of high and moderate impact variants, considering only variants for which population-specific minor allele frequencies (MAFs) were likely <0.1% (Methods). Both analyses were performed for a recessive and a dominant model of inheritance. Given that copy-number variants (CNVs), at least in the

**Fig. 1 | Phenotypic definition of the cohort, diagnostic yield, identification of predictive factors, and gene-burden analysis. a** Phenotypic similarities between patients, encoded by their HPO terms, were visualised using uniform manifold approximation and projection (UMAP). The clinical phenotype space was first defined by all OMIM diseases using their HPO annotations (grey dots). Each patient analysed by exome sequencing (ES) or genome sequencing (GS) is shown as a coloured dot, with colours indicating movement disorders subgroups. Triangles denote patients carrying *CD99L2* loss-of-function variants. **b** The proportion of individuals with firm diagnoses (likely pathogenic or pathogenic variants) was slightly higher in the first-line GS (GS first) cohort (25.2%, *n* = 214) compared to the first-line ES (ES first) cohort (21.96%, *n* = 1557). The lowest diagnostic yield was seen in the second-line GS cohort after prior ES and optional fragment analysis (FA) (10.5%, *n* = 200). **c** Distribution of variant types among clinically relevant likely pathogenic and pathogenic variants in the ES first cohort (*n* = 342), GS first cohort (*n* = 54), and GS with prior ES ± FA (*n* = 21). **d** Visualisation of least absolute shrinkage and selection operator (LASSO) coefficients across different lambda values. **e** LASSO coefficients at minimum lambda. **f** Manhattan plot of *P* values from the gene-burden analysis assessing associations between high-impact variants and movement disorders. The x-axis indicates chromosomal positions, and the y-axis shows the $-\log_{10}(P)$ values for the two-tailed association test per gene (log scale). For each gene, the lowest *P* value among three tested models (dominant, recessive, and recessive plus CNVs) is shown. The red dotted line marks the significance threshold ($P = 4.9 \times 10^{-6}$, Bonferroni correction for 10,165 genes affected by high-impact variation in the MD cohort), the dark red line corresponds to $P = 0.0025$. Labels in light red indicate established disease genes associated with neurological phenotypes with $P < 4.9 \times 10^{-6}$, dark red labels genes with $P < 0.0025$. The blue label marks the newly prioritised disease gene *CD99L2*. Source data are provided as a Source Data file.

homozygous state, can be detected with a high sensitivity and specificity in genomes and in uniformly processed exome datasets, we extended the recessive stringent model to include high-quality homozygous exon deletions. Of the top three associations identified, two (*SPAST* and *SACS*) were already known according to OMIM and PanelApp, and one (*CD99L2*) was previously unidentified. Notably, of the top 20 ranked genes, 11 (*ABCD1, KIF1C, POLR3A, REEP1, SACS, SIL1, SLC44A4, SPAST, SPG11, STUB1, UCHL1*) have previously been linked to neurological phenotypes, confirming the utility of the chosen approach (Fig. 1f, an extended prioritisation list is provided in Supplementary Data 1). To prioritise candidates for further investigation, we evaluated supportive evidence from co-segregation and performed a comprehensive review of the literature for each gene to assess whether an association was supported by biological function or previously known disease associations for related genes. Overall, *CD99L2* remained the most compelling candidate and underwent further investigation through replication in other rare disease cohorts and in-depth functional studies.

**Loss-of-function variants in *CD99L2* cause X-linked spastic ataxia.** Subsequent to the seven index cases identified in the initial analysis, an additional eight index cases were detected during routine genome-based diagnostic work-up, one by an extended search of in-house research datasets, and four families via the matchmaking functionality of the RD-Connect Genome-Phenome Analysis Platform (GPAP)[10] from the Solve-RD rare neurological diseases cohort[26]. Of the total 25 affected male individuals from 20 families (Fig. 2a–k and Supplementary Fig. 7a–i), 12 had received a primary diagnosis of hereditary spastic paraplegia (HSP), ten of spastic ataxia, and three of cerebellar ataxia. Age at onset ranged from 10 to 68 years (median: 47 years; inter-quartile range (IQR): 13 years), and the main presenting features were gait disturbances and spasticity (22/25; 88%), predominantly affecting the lower limbs. During the course of the disease, 41.6% (5/12) of the patients with HSP showed signs of ataxia. Dysarthria (21/25; 84%), oculomotor abnormalities (14/25; 56%), sensory deficits (13/24; 54%), and dysphagia (10/25; 40%) were additional common clinical features (Supplementary Fig. 6 and Supplementary Data 2; see Supplementary information: Case reports *CD99L2* families). Nine patients were dependent on a walking aid at a median age of 59 years (range 44–62 years; IQR: 7.25 years). No specific pathologies were detected in the brain MRI examinations performed in 19 probands; only in two cases cerebellar atrophy was observed. Female carriers showed no specific phenotype (Table 1, Supplementary Data 2 and Supplementary Fig. 6).

In summary, the identification of 11 different disease alleles (Fig. 2l), most of which predict a loss of CD99L2 function, in a total of 25 affected members of 20 families with overlapping phenotypes, provides strong evidence that CD99L2 is implicated in X-linked spastic ataxia.

**Consequences of identified *CD99L2* variants on transcript and protein levels.** To further support the functional relevance of identified variants, we conducted transcriptome sequencing (TS) and western blot studies on available patients' biospecimens. These included analyses of PAXgene Blood RNA Tubes, representing all identified variants, and fibroblasts from affected individuals carrying six different variants (Fig. 3 and Supplementary Fig. 7j–o). For all PTVs studied, *CD99L2* mRNA levels were significantly reduced compared to controls, in line with nonsense-mediated decay (NMD) of mutant transcripts (Fig. 3b, c). When quantified as read counts, *CD99L2* mRNA levels of individuals with other mutations, including predicted splice-site alterations and exon deletions, were not significantly altered. However, no wild-type transcripts were observed, only transcripts indicating exon skipping, intron retention, and/or variable use of alternative splice sites (Fig. 3a), as well as a potential upregulation of the shorter transcript ENST00000418547.2 (Ensembl ID) (Fig. 3a; transcript c) of variant c.655+1 G > A. Apart from the exon 8/9 deletion, which results in an in-frame transcript missing the entire transmembrane domain, all other aberrant mRNAs predict mainly prematurely truncated proteins. The use of alternative splice sites as shown in Fig. 3a; transcript b) of variant c.535 G > C and c.655+1 G > A was only seen in a smaller subset of reads.

To analyse the consequences of the identified *CD99L2* variants at the protein level, we performed western blotting of fibroblast-derived protein extracts using antibodies directed against either the N-terminus (targeting residues 27–117) or the C-terminus (targeting residues 209–254) of the protein (Fig. 3d–i and Supplementary Fig. 7j–o). In controls, a strong band was detected between 37 and 50 kDa using the N-terminal antibody– higher than the predicted molecular weight of 28 kDa for full-length CD99L2. However, the specificity of this signal was confirmed by parallel detection using the C-terminal antibody.

In patients carrying PTVs, CD99L2-specific bands were entirely absent (Fig. 3d–f and Supplementary Fig. 7j–l). In contrast, patients with splice-site alterations or a deletion of exons 8/9 showed faint bands of lower molecular weight when probed with the N-terminal antibody (Fig. 3g–i) and–exclusively in the case of the exon 8/9 deletion–also with the C-terminal antibody (Supplementary Fig. 7m–o). These may represent internally or C-terminally truncated CD99L2 protein forms, likely corresponding to translation products from the aberrant transcripts detected in the transcriptome analysis (Fig. 3a).

In conclusion, all *CD99L2* variants investigated led to a near-complete loss of full-length CD99L2, supporting a loss-of-function as the likely pathomechanism.

**Characterisation of CD99L2 function in health and disease.** Similar to CD99, CD99L2 has been postulated to play a role in leucocyte transmigration through endothelial cell-cell contacts[21]. However, apart from an association with autism spectrum disorder[22] and cerebral palsy[23], it has not yet been linked to neuronal phenotypes in humans, while a study in mice has suggested that it functions as a synaptic cell adhesion molecule important for regulating neuronal activation[27].

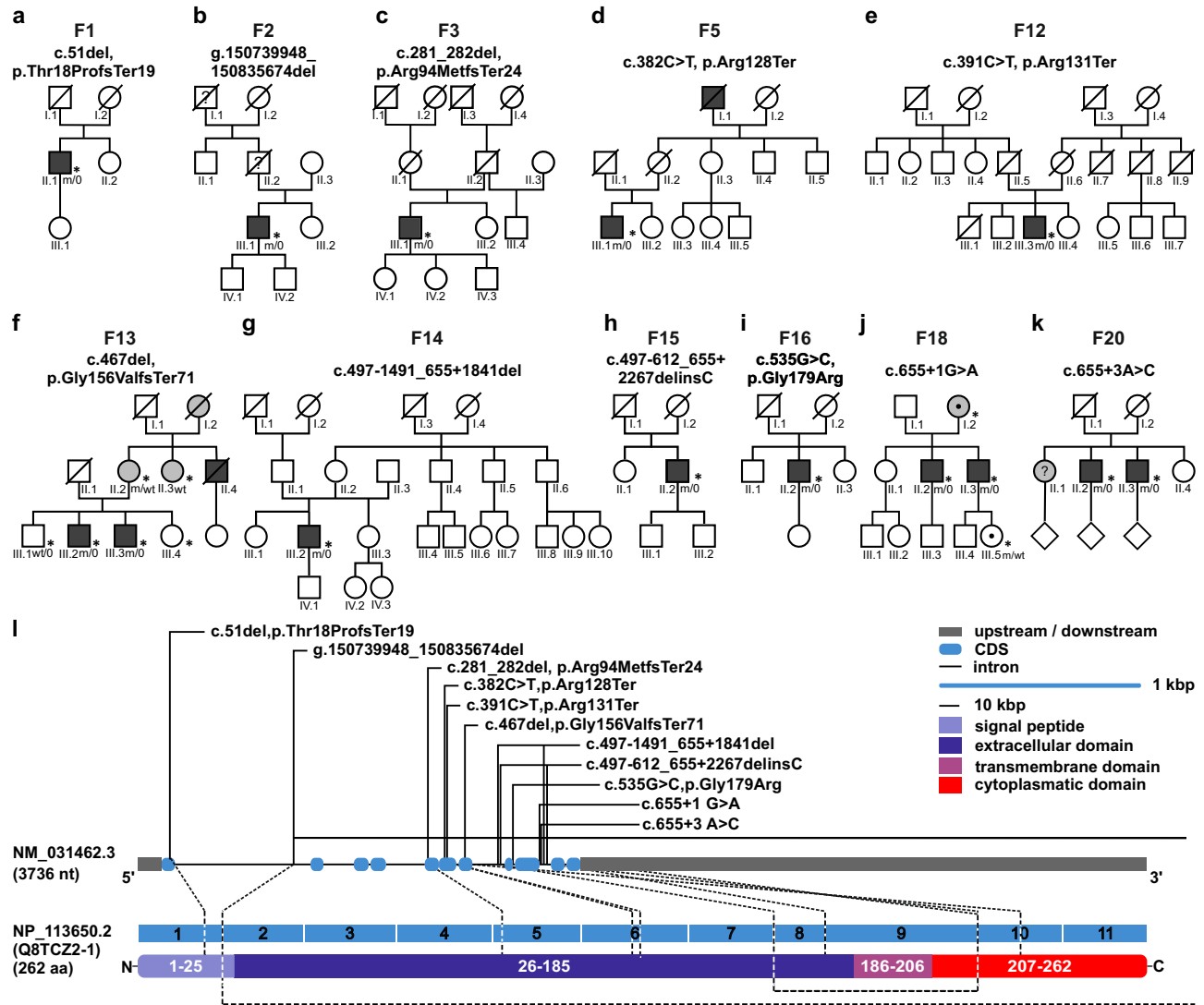

**Fig. 2 | *CD99L2* mutation status and distribution of identified variants.**
**a–k** Pedigrees of the investigated families carrying functionally relevant variants in *CD99L2*. The observed inheritance patterns are consistent with an X-linked model, showing full penetrance in males and variable penetrance and expressivity in female carriers. Asterisks (*) indicate fully characterised patients. **l**, Gene structure of *CD99L2* depicting known domains of the encoded protein, the localisation of identified variants. Intronic regions are not drawn to scale. Source data are provided as a Source Data file.

**CD99L2 primarily exists as a ubiquitinated protein and is degraded via autophagy and proteasomes.** To investigate the discrepancy between the calculated (~28 kDa) and observed (37–50 kDa) molecular weight of CD99L2 in control fibroblasts (Fig. 3g–i and Supplementary Fig. 7j–o), we first replicated these findings in 293T cells and SH-SY5Y neuroblastoma cells, both at baseline and upon overexpression of a CD99L2 construct. Only upon overexpression, at levels several-fold higher than the endogenous levels, we were able to detect the apparently unmodified full-length form of CD99L2, along with a potential N-terminal fragment, while the modified form represented the predominant species (Fig. 4a and Supplementary Fig. 8a–d). We then explored possible sources of posttranslational modification (PTM) in 293T cells using pharmacological strategies. While previous studies suggested *O*-glycosylation as the primary PTM of CD99L2 responsible for the molecular weight shift[28], we ruled out *O*-GlcNAcylation as the main modification (Supplementary Fig. 8e–g). On the other hand, inhibition of proteasomes and autophagy (Fig. 4b)—two protein degradation pathways that rely on substrate ubiquitination as a process-specific marker—led to a marked increase in both the modified and unmodified full-length forms of CD99L2 (Fig. 4c, d). Notably, proteasomal inhibition by

MG132 was the most effective in elevating CD99L2 levels, leading to an ~12-fold increase of full-length CD99L2 compared to a five-fold increase upon autophagy inhibition using bafilomycin A1 (BafA1) (Fig. 4d). Moreover, while BafA1 treatment increased levels of the unmodified full-length CD99L2, the levels of the modified, potentially ubiquitinated form declined, likely reflecting of a compensatory upregulation of proteasomal activity in response to autophagy inhibition.

To confirm that CD99L2 undergoes ubiquitination, we performed immunoprecipitation experiments targeting ubiquitinated substrate proteins in CD99L2-expressing 293T cells, which confirmed the presence of ubiquitinated CD99L2 at the expected molecular size (Fig. 4e, f and Supplementary Fig. 8h–k). We validated these findings by mutating all 14 lysine residues in CD99L2 (Fig. 4g) to arginine (CD99L2$_{K0}$) to prevent canonical ubiquitination, which successfully resulted in a shift from the higher molecular weight form of CD99L2 to its unmodified full-length form (Fig. 4h, i). While this effect was pronounced, it did not completely eliminate the presence of modified CD99L2, which can be attributed to residual ubiquitination within the protein tag, non-canonical ubiquitination, or the contribution of previously reported glycosylation.

**Table 1 | Characteristics of patients with *CD99L2*-related spastic ataxia\***

| Characteristic | Male | Female† |
|---|---|---|
| No. | 25 | 4 |
| Mean age at onset (years) | 43 ± 11 | 68 ± 2 |
| Signs and symptoms – no./total no. (%) | | |
| Spasticity | 22/25 (88) | 1/4 (25) |
| Dysarthria | 21/25 (84) | 0/4 (0) |
| Ataxia | 18/25 (72) | 1/4 (25) |
| Oculomotor abnormalities | 14/24 (58) | 1/4 (25) |
| Sensory deficits | 13/24 (54) | 1/4 (25) |
| Pyramidal signs | 13/24 (54) | 0/4 (0) |
| Bladder dysfunction | 12/24 (52) | 1/4 (25) |
| Dysphagia | 10/25 (40) | 1/4 (25 |
| Contractures | 1/22 (4) | 0/4 (0) |
| Cerebellar atrophy on MRI – no./total no. (%) | 2/19 (11) | 0/0 |

*Plus-minus values means ± SD. Clinical data from 29 individuals with *CD99L2*-related neurological phenotypes were included.
†Data on age at onset of putative presenting signs were missing for two patients.

Protein ubiquitination is regulated by a dynamic interplay between substrate-specific E3 ubiquitin ligases (E3s) and deubiquitinases (DUBs)[29]. As the CD99L2-specific E3s and DUBs remain unidentified, we co-transfected the DUB ataxin-3, a polyglutamine protein associated with spinocerebellar ataxia type 3[30], with CD99L2 in 293T cells to analyse its effects on CD99L2 ubiquitination. As expected, ataxin-3 over-expression led to a decrease in lysine-48-linked polyubiquitin chains. Interestingly, we also observed a reduction in ubiquitinated CD99L2, accompanied by increased levels of the unmodified full-length protein (Supplementary Fig. 8l–n), further supporting our findings and suggesting a potential pathomechanistic link.

**CD99L2 binds CAPN1 and triggers calpain activation.** Previous affinity capture-based mass spectrometry identified CD99L2 as an inter-actor of calpain-1 (CAPN1), a calcium-activated cysteine protease involved in neuronal plasticity and implicated in recessive spastic paraplegia and cerebellar ataxia[19,20]. To validate these findings, we performed co-immunoprecipitation (co-IP) experiments using two differently tagged CD99L2 constructs overexpressed in 293T cells. Western blot analysis confirmed that CD99L2 directly interacts with CAPN1 and apparently co-precipitates its regulatory small subunit CSS1 (Fig. 5a, b and Supplementary Fig. 9a–d). A potential weak binding to calpain-2 (CAPN2) was also observed (Extended Data Fig. 10d), whereas no interaction was detected with the non-classical calpain-10 (CAPN10), which structurally differs from CAPN1 by lacking the C-terminal calcium-binding penta-EF-hand domain—and thereby the ability to interact with CSS1[31]—or with the endogenous proteinaceous inhibitor calpastatin (CAST).

In our co-IP experiments, we observed increased autoproteolysis of CAPN1, reduced levels of the endogenous proteinaceous inhibitor calpastatin (CAST), and a fragmentation-related decrease in full-length CSS1, suggesting activation of the calpain system upon CD99L2 transfection. To confirm this, we repeated CD99L2 overexpression in 293T and SH-SY5Y cell lines and quantitatively analysed key protein markers, including the known calpain substrate α-spectrin, whose calpain-mediated proteolysis generates a characteristic ~150 kDa fragment. All analysed proteins exhibited enhanced proteolytic processing in both cell models (Fig. 5c, d and Supplementary Fig. 9e–h), corroborating CD99L2-induced activation of calpains.

The calpain dependency of this effect was further validated through inhibition experiments. Both co-overexpression of CAST and a 24-h treatment with calpain inhibitor III (CI III) were able to counteract CD99L2-dependent calpain activation, as demonstrated by prevention of α-spectrin cleavage (Fig. 5e–h and Supplementary Fig. 10). Notably, CI III treatment showed stronger effects than CAST modulation. Interestingly, CAST overexpression and CI III administration led to increased levels of full-length CD99L2, along with a reduction of the N-terminal CD99L2 fragment, indicating that CD99L2 itself is a calpain substrate. In conclusion, we show that CD99L2 specifically interacts with CAPN1 and activates the calpain system when overexpressed.

**CD99L2 is a calpain substrate and is cleaved at its cytoplasmic C-terminus.** Both genetic and pharmacological inhibition of calpains increased levels of full-length CD99L2 while reducing levels of the observed N-terminal fragment. To confirm that CD99L2 is indeed a substrate of calpain-mediated proteolysis and that the fragment is a product of this process, we first performed esiRNA-mediated knockdown of *CAPN1* and *CAPN2* in 293T cells. This analysis revealed that only *CAPN1* depletion affected endogenously expressed CD99L2. Interestingly, *CAPN1* knockdown increased the upper, high-molecular-weight fraction of ubiquitinated CD99L2, while lower bands disappeared, suggesting that both full-length and cleaved CD99L2 exist in ubiquitinated forms (Fig. 6a, b and Supplementary Fig. 11a).

To investigate this further, we analysed fibroblasts from a patient with *CAPN1*-related spastic paraplegia type 76 (SPG76)[32], which exhibit loss of the CAPN1 protein. In comparison to a control line, the SPG76 fibroblasts showed increased levels of CAST and reduced α-spectrin cleavage (Fig. 6c, e). Notably, SPG76 fibroblasts also showed reduced fragmentation of ubiquitinated CD99L2, confirming CAPN1 as a key proteolytic regulator of CD99L2 (Fig. 6d, f).

Since effects on the unmodified full-length CD99L2 and its N-terminal fragment are difficult to assess at endogenous expression levels, we next analysed calpain-mediated cleavage in 293T cells overexpressing CD99L2. When incubating protein extracts with exogenous CAPN1, we observed rapid degradation of both full-length and ubiquitinated CD99L2, along with an accumulation of N-terminal ubiquitinated fragments (Fig. 6g, h). Using a less acute, cell-based approach, we treated transfected 293T cells with the calcium iono-phore ionomycin to induce calpain activation. This resulted in a marked reduction of both ubiquitinated and unmodified full-length CD99L2, accompanied by an increase in the N-terminal fragment (Fig. 6i, j and Supplementary Fig. 11b, c).

In silico prediction tools suggested that the observed N-terminal fragment likely results from a cleavage site within the calpain-accessible cytoplasmic domain of CD99L2, located around serine 250 (Fig. 6k, l)[33]. To confirm the presence of this calpain cleavage site, we employed a mutagenesis strategy previously developed for the ataxin-3 protein[34]. Substitution of residues 247–250 with tryptophan rendered CD99L2 resistant to calpain-mediated fragmentation, effectively blocking formation of the N-terminal breakdown product (Fig. 6l, m). These results demonstrate that CD99L2 is a calpain substrate with a specific cleavage site located at serine 250.

**Ubiquitination of CD99L2 modulates CAPN1 binding and calpain activation.** After demonstrating that CD99L2 directly binds to CAPN1 and induces calpain system activation, we explored whether its ubiquitination capacity contributes to this function. To investigate this, we overexpressed both myc-tagged wild-type CD99L2 and its lysine-free variant CD99L2$_{KO}$ in 293T cells. Fluorescence microscopy revealed no differences in subcellular localisation of the two CD99L2 variants, including their limited colocalization with CAPN1 at membrane-distal contact sites, indicating that lysine residues—and their ubiquitination—do not alter CD99L2 targeting to the plasma membrane (Fig. 7a).

To assess changes in protein interaction, we performed co-IP via the myc tag of the CD99L2 constructs. Western blot analysis revealed a markedly increased binding affinity of CD99L2$_{KO}$ to CAPN1 compared

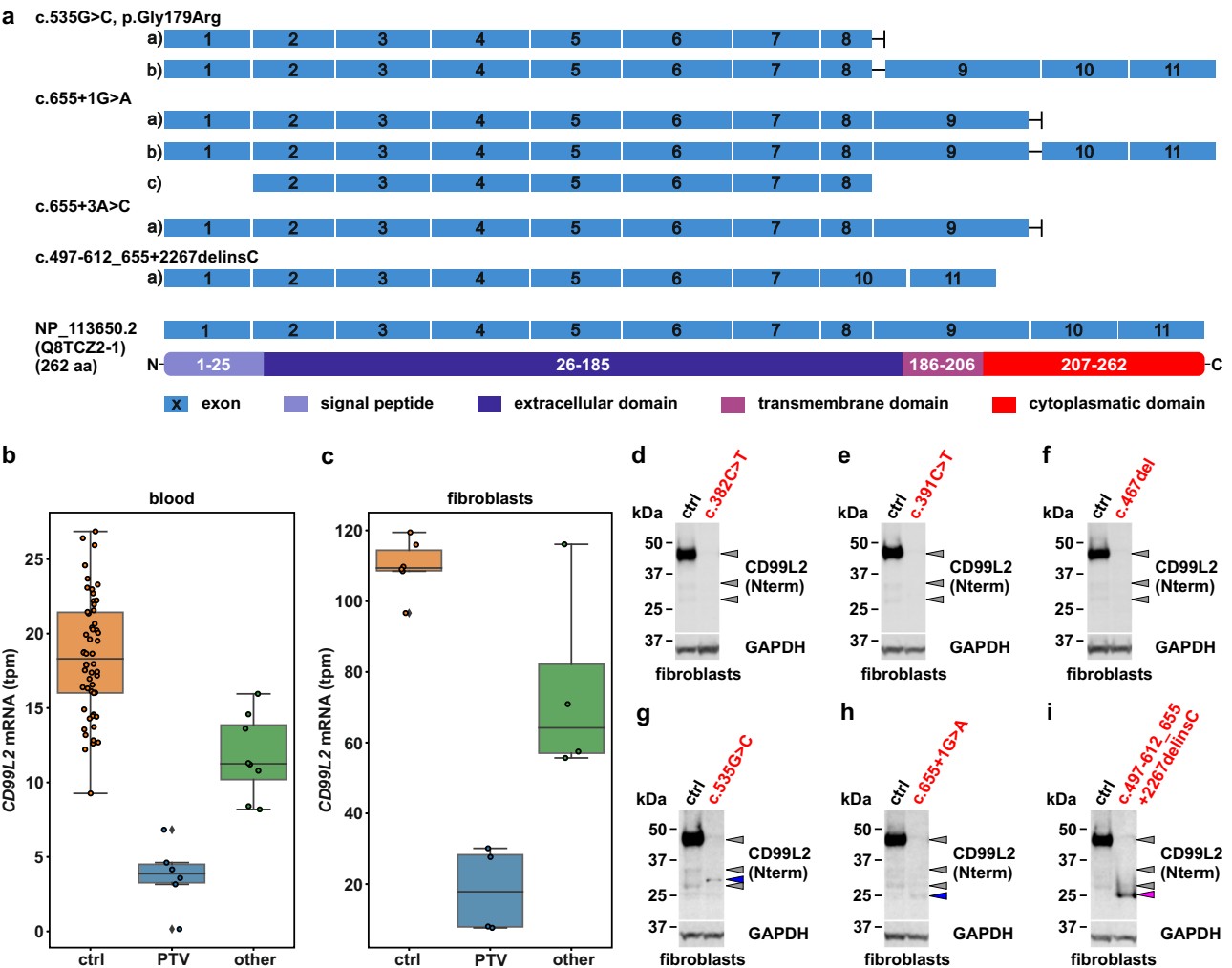

**Fig. 3 | Molecular consequences of *CD99L2* variants. a** Transcript-level consequences of *CD99L2* variants. Schematic representation of splicing effects caused by variants c.535 G > C, p.Gly179Arg, c.655+1 G > A, c.655+3 A > C and c.497-612_655+2267delinsC, as observed at the transcriptome level. Transcript c of variant c.655+1 G > A corresponds to ENST00000418547 (Ensembl transcript ID). For comparison, the exon and protein structure of *CD99L2* are shown below (NCBI Reference Sequence: NP_113650.2; UniProt ID: Q8TCZ2-1). **b** RNA expression levels of *CD99L2* (transcripts per million, tpm) in blood from 14 index patients grouped into the categories protein-truncating variants (PTV, blue; p.Arg128Ter, p.Arg131-Ter, p.Gly156ValfsTer71, deletion including exons 2-11; min 0.15, lower whisker 3.16, lower quartile 3.27, median 3.87, upper quartile 4.5, upper whisker 4.61, max 6.82) and other mutations (green; c.655+1 G > A, c.655+3 A > C, p.Gly179Arg, deletion of exons 8/9; min/lower whisker 8.19, lower quartile 10.19, median 11.26, upper quartile 13.85, upper whisker/max 15.95), compared to 54 male controls (orange;

min/lower whisker 9.26, lower quartile 16.0, median 18.30, upper quartile 21.43, upper whisker/max 26.86). **c** RNA expression of *CD99L2* (tpm) in fibroblasts from eight index patients grouped into the categories PTV (blue; p.Arg128Ter, p.Arg131Ter, p.Gly156ValfsTer71; min/lower whisker 7.64, lower quartile 7.93, median 17.85, upper quartile 28.28, upper whisker/max 30.11) and other mutations (green; c.655+1 G > A, deletion of exons 8/9; min/lower whisker 55.67, lower quartile 57.02, median 64.17, upper quartile 82.18, upper whisker/max 116.08), compared to six male controls (orange; min 96.65, lower whisker 108.46, lower quartile 108.61, median 109.37, upper quartile 114.40, upper whisker/max 119.44). **d–i** Western blot of CD99L2 in primary fibroblasts from healthy controls (ctrl) and patients (indicated in red script), using an N-terminal CD99L2 antibody. Grey arrowheads indicate CD99L2-specific bands in controls, blue arrowheads C-terminally truncated CD99L2 (**g**, **h**), and the purple arrowhead an internally truncated form (**i**). GAPDH served as loading control. Source data are provided as a Source Data file.

to wild-type CD99L2 (Fig. 7b). Notably, the absence of lysine residues in CD99L2 did not enhance its weak binding to CAPN2.

Next, we assessed the impact of lysine residue loss on CD99L2's ability to activate calpains. Western blot analysis of calpain system markers− including CAPN1 and CAST levels, as well as α-spectrin and CSS1 cleavage−indicated a further amplification of proteolytic activation in cells expressing CD99L2$_{KO}$ (Fig. 7c, d).

Together, these findings suggest that the ubiquitination of CD99L2 modulates its interaction with CAPN1 and dampens calpain activation, potentially serving as a regulatory mechanism to prevent excessive and harmful calpain activity within cells.

**CD99L2 domain truncation compromises intracellular localisation, calpain interaction and calpain activation.** We next aimed to

characterise the functional role of individual CD99L2 domains by overexpressing myc-tagged CD99L2 truncation constructs containing either the extracellular region, including the N-terminal signal peptide (CD99L2$_{1–185}$), or the cytoplasmic portion (CD99L2$_{207–262}$) (Fig. 8a). Our analysis underscored the importance of CD99L2's structural integrity: while all constructs showed limited colocalization with CAPN1, CD99L2$_{1–185}$ displayed a compartmentalised cytoplasmic pattern reminiscent of the endoplasmic reticulum (ER), which was confirmed via coexpression with a DsRed-tagged ER reporter. In contrast, CD99L2$_{207–262}$ exhibited a more diffuse distribution in the cytoplasm and, to a lower extent, the nucleus. Full-length CD99L2, as expected, localised to the plasma membrane (Fig. 8b and Supplementary Fig. 12a, b).

Importantly, while not disturbing CD99L2 protein levels, the truncation of CD99L2 abolished its ubiquitination (Fig. 8c) and

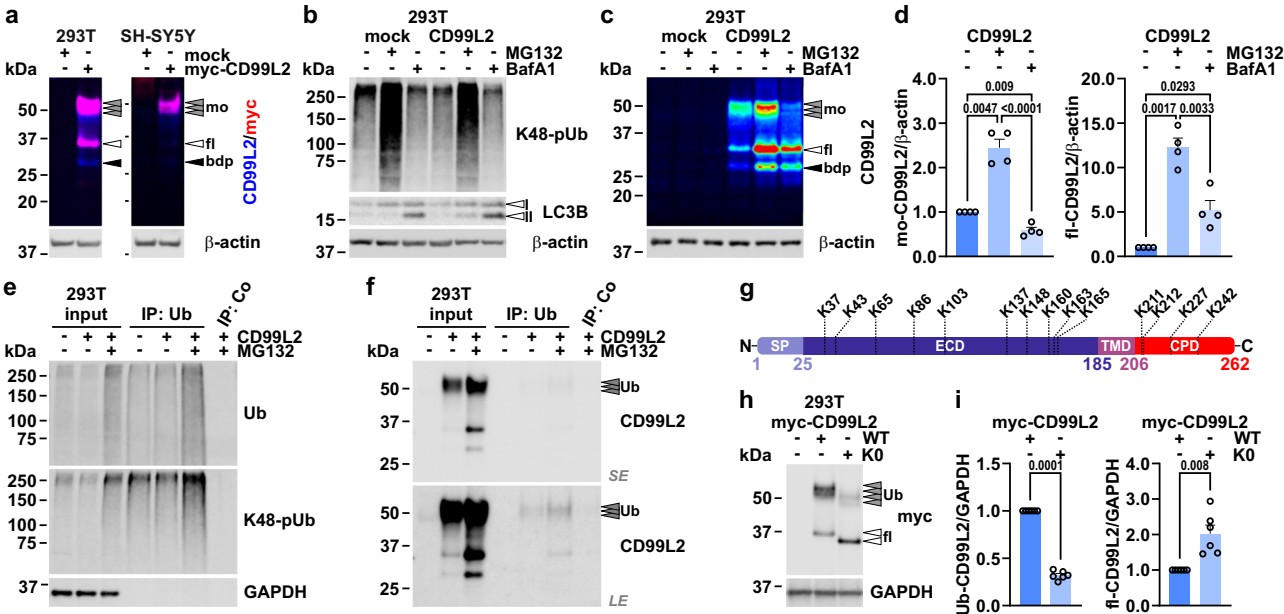

**Fig. 4 | CD99L2 is ubiquitinated, and degraded via autophagy and proteasomes. a** Western blot of myc-His$_6$-CD99L2 in 293T and SH-SY5Y cells using CD99L2- and myc tag-specific antibodies. β-actin served as loading control. Grey arrowheads indicate modified full-length (fl$_{mo}$) CD99L2, white arrowheads unmodified full-length (fl) CD99L2, and black arrowheads a breakdown product (bdp). **b, c** Western blots of K48-linked polyubiquitin (K48-polyUb), LC3B-I/II, and CD99L2 in 293T cells expressing myc-His$_6$-CD99L2 after proteasomal (MG132) or autophagosomal (bafilomycin A1, BafA1) inhibition. β-actin served as loading control. **d** Quantification of modified and full-length CD99L2 after treatment. $n = 4$ biologically independent experiments. One-sample $t$- or Student's $t$-tests. **e, f** Ubiquitination analysis of myc-His$_6$-CD99L2 by ubiquitin-directed immunoprecipitation (IP: Ub) in MG132-treated cells. Total- and K48-linked ubiquitin was detected (**e**). CD99L2 was detected with an N-terminal antibody (**f**). GAPDH served as loading control. Grey arrowheads indicate ubiquitinated (Ub) CD99L2. *LE*, long exposure; *SE*, short exposure. **g** Schematic of lysine residue distribution within CD99L2. SP signal peptide, ECD extracellular domain, TMD transmembrane domain, CPD cytoplasmic domain. **h** Western blot of wild-type (WT) and lysine-free (K0) CD99L2 variants (14 lysine residues mutated to arginine) in 293T cells detected with a myc tag-specific antibody. GAPDH served as loading control. **i** Quantification of WT and K0 CD99L2. $n = 6$ biologically independent experiments. One-sample $t$-test. Bars represent mean ± s.e.m. All $t$-tests are two-tailed. $P$ values are shown in the graphs. Source data are provided as a Source Data file.

impaired calpain activation due to disrupted CAPN1 binding (Fig. 8d–g and Supplementary Fig. 12c–e), highlighting the necessity of membrane anchoring for proper function.

Given the previously observed bimodal degradation of full-length CD99L2, we next explored whether specific domains influence its proteostatic fate. Notably, CD99L2$_{1–185}$ accumulated upon autophagy inhibition, while CD99L2$_{207–262}$ levels increased following proteasome inhibition (Fig. 8h–j and Supplementary Fig. 12f–i). These findings suggest that CD99L2's bimodal degradation is domain-dependent, with the N-terminal region driving its autophagosomal clearance and the C-terminal domain directing proteasomal degradation.

**CD99L2 patient fibroblasts show transcriptional dysregulation of genes associated with neuronal function.** After functionally dissecting the CD99L2 protein, we next analysed the downstream consequences of CD99L2 variants to better explain the neurological phenotype observed in affected individuals. To this end, we performed transcriptomic profiling of fibroblasts from patients carrying PTVs, splice-site variants, or deletions, compared to fibroblasts from unaffected matched controls. RNA sequencing identified 515 significantly differentially expressed genes (DEGs), including 191 downregulated and 324 upregulated transcripts (Fig. 9a). Since the physical interaction of CD99L2 and the calpain system occurs at the protein level, we specifically assessed potential transcriptional dysregulations of related genes. However, no significant expression changes were detected for *CAPN1, CAPN2, CAPNS1* (encoding CSS1), *CAST* or *SPTAN1* (encoding the calpain substrate α-spectrin) (Supplementary Fig. 13a). Furthermore, we analysed protein levels of CAPN1 and CAST, as well as calpain-mediated cleavage of α-spectrin, in fibroblasts derived from CD99L2 patients and healthy controls. While no significant alterations in CAPN1 or CAST levels were detected, spectrin cleavage was reduced

in patient fibroblasts, consistent with the absence of the CAPN1 activator CD99L2 (Supplementary Fig. 13b, c).

To identify functional themes among the DEGs, we performed STRING-based gene ontology (GO) enrichment analysis[35]. Among the top ten enriched biological processes, eight were directly linked to neuronal function. Similarly, six of the top ten enriched cellular compartments were related to neurons, particularly the synapse (Fig. 9b, c). Beyond the general GO term "cell junction", the strongest enrichment was observed for "synapse," which encompassed 55 DEGs showing marked dysregulation and a high degree of functional interaction (Fig. 9d, e and Supplementary Fig. 13d).

Although fibroblasts are not the model of choice for directly studying neuron-specific pathology, the consistent dysregulation of neuron- and synapse-related genes highlights the likely impact of *CD99L2* variants on neuronal function and degeneration. In light of previous findings that loss of CAPN1 impairs synaptic function and triggers cerebellar ataxia[36], and given that CD99L2 may regulate synapse development[27], our data suggests that perturbed CD99L2 function could exert similarly detrimental effects in the human brain.

## Discussion

It is recognised that rare genetic diseases are arguably the largest disease area of unmet need, with a real paucity of therapeutic options for affected individuals. Pushing the boundaries to improve diagnosis and ultimately patient outcomes is a key priority with manifold socioeconomic implications. We and others have shown that expanding the diagnostic target 'beyond-the-exome' by GS and concomitant implementation of bioinformatic pipelines tailored to detect additional types of genomic variation can improve the diagnostic yield by 5–20%[25,37,38]. In the field of MDs, we highlight the diagnostic potential of data analysis for rare disease-causing REs. Diagnostic-grade

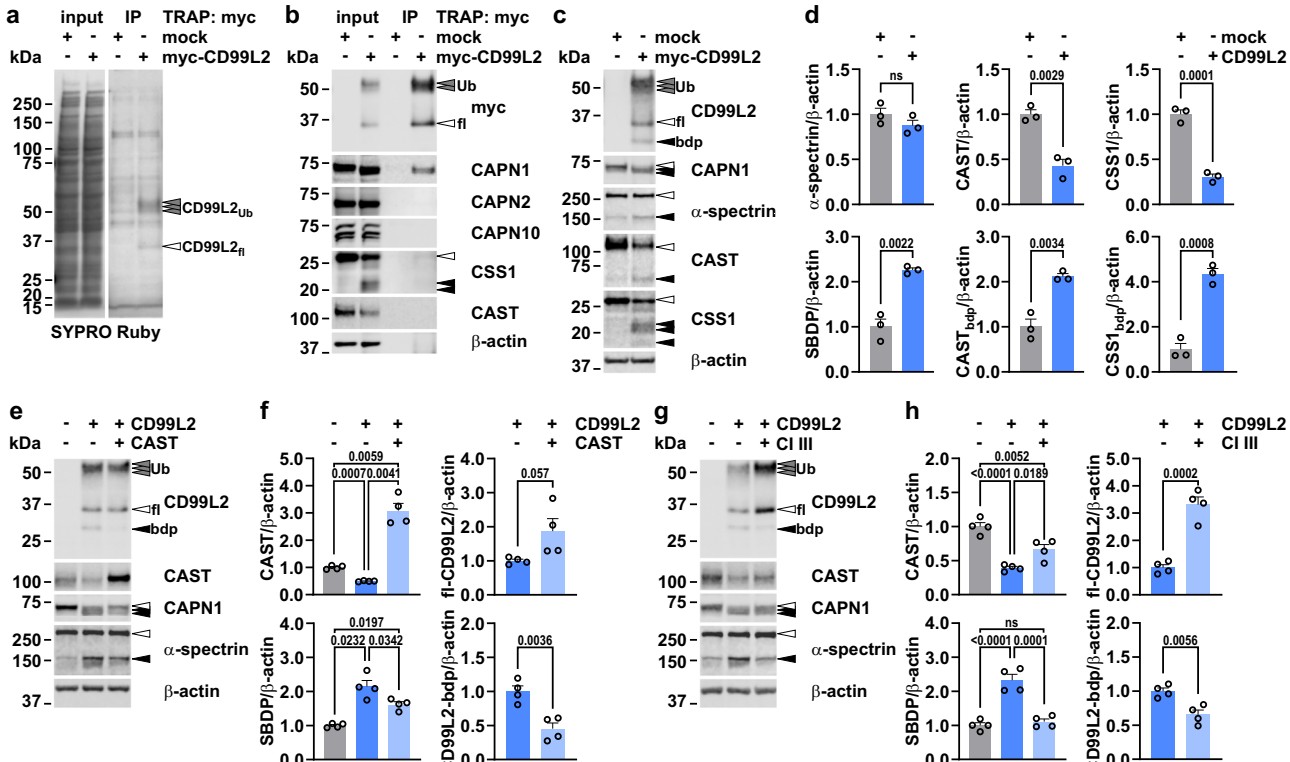

**Fig. 5 | CD99L2 functions as an interactor and activator of CAPN1. a, b** Co-immunoprecipitation (myc TRAP) of myc-His$_6$-CD99L2 in 293T cells. Total protein was visualised using SYPRO Ruby. CAPN1, CAPN2, CAPN10, CSS1 and CAST were detected. CD99L2 was visualised using a myc tag antibody. β-actin served as loading control. Grey arrowheads show ubiquitinated (Ub)-CD99L2, white arrowheads unmodified full-length (fl)-proteins, and black arrowheads protein fragments. **c** Western blot of myc-His$_6$-CD99L2, CAPN1, α-spectrin, CAST and CSS1 in 293T cells. β-actin served as loading control. bdp, CD99L2 breakdown product. **d** Quantification of full-length und calpain-cleaved proteins detected in (**c**). SBDP, calpain-cleaved α-spectrin. $n = 3$ biologically independent experiments. Student's $t$-

test. **e** Western blot of CD99L2, CAST, CAPN1, and α-spectrin in 293T cells co-expressing myc-His$_6$-CD99L2 and CAST. β-actin served as loading control. **f** Quantification of proteins detected in (**e**). $n = 4$ biologically independent experiments. One-way ANOVA with Tukey's post hoc test or Student's $t$-test. **g** Western blot of CD99L2, CAST, CAPN1 and α-spectrin in 293T cells expressing myc-His$_6$-CD99L2 upon calpain inhibitor III (CI III) treatment. β-actin served as loading control. **h** Quantification of markers detected in (**i**). $n = 4$ biologically independent experiments. One-way ANOVA with Tukey's post hoc test or Student's $t$-test. Bars represent mean ± s.e.m. All $t$-tests are two-tailed. $P$ values are shown in the graphs; ns not significant. Source data are provided as a Source Data file.

implementation of long-read GS with more complete reference genomes is expected to further address the difficulty of accurate detection of complex SVs and characterisation of REs. With the ability to identify epigenetic modifications, long-read GS has the potential to constitute a one-test solution that informs on most clinically relevant genetic variation. However, while technological improvements will continue to enhance diagnostic sensitivity, the availability of harmonised, representative population-scale datasets and the development of statistical methodology will be crucial to interpret the new data. To this end, the need for healthcare standards and infrastructures for rare diseases is widely accepted, and a national sequencing programme and research environment has proven a powerful framework[3–5]. These (inter)national endeavours are paving the way for the multi-ethnic exploration of SVs, rare variation in non-coding genes and regulatory elements of the genome, and additional models of inheritance such as polygenic effects and reduced penetrance. In this context, it is widely accepted that consented genetic data generated as part of routine diagnostic practice should be shared with the research community to drive discovery of novel disorders and guide the clinical reporting process.

The German concept for diagnostic ES[24] has provided valuable guidance for the development of the National Strategy for Genomic Medicine (§64e of Social Code Book V), which creates the legal basis for the examination and treatment of patients with rare diseases and cancer using GS.[39] added clinical value of multidisciplinary case conferences as a prerequisite for broad diagnostic sequencing approaches, and in particular the scalability of such patient care pathways, remains

to be evaluated. Nevertheless, they certainly serve educational aspects in academia and are prototypical examples of translating expertise shared between clinicians, data scientists, and basic researchers into diagnoses and a better understanding of disease biology as a basis for precision care concepts and clinical intervention.

In this pilot study, we have demonstrated the clinical feasibility of streamlined data management in the setting of a single tertiary care centre linked to international RDs research environments. We exemplify the conceptual added clinical value in a particularly large and likely genetically heterogeneous cohort, using a simple statistical test to contribute to the resolution of an unknown rare disease aetiology, namely *CD99L2*-associated spastic ataxia. While previous studies focused on single-nucleotide variants and small insertions and deletions in the coding regions, we illustrate that expanding the mutational spectrum to include high-quality CNV calls can support the identification of genome-wide significant discoveries. Of note, while careful partitioning of heterogeneous case sets using individual-level HPO terms has been suggested to boost power, our study suggests that at least in MDs with broad clinical heterogeneity, subgroup collision might be beneficial. Both, the variability in the clinical assignment to main diagnoses, as well as the heterogeneity in the disease-causing variant types, could possibly explain why this presumably more frequent disease gene has remained undiscovered so far.

Our functional investigations revealed that the membrane protein CD99L2, in addition to its known role in leucocyte transmigration[21], acts as a potent and specific interactor and activator of the calcium-activated

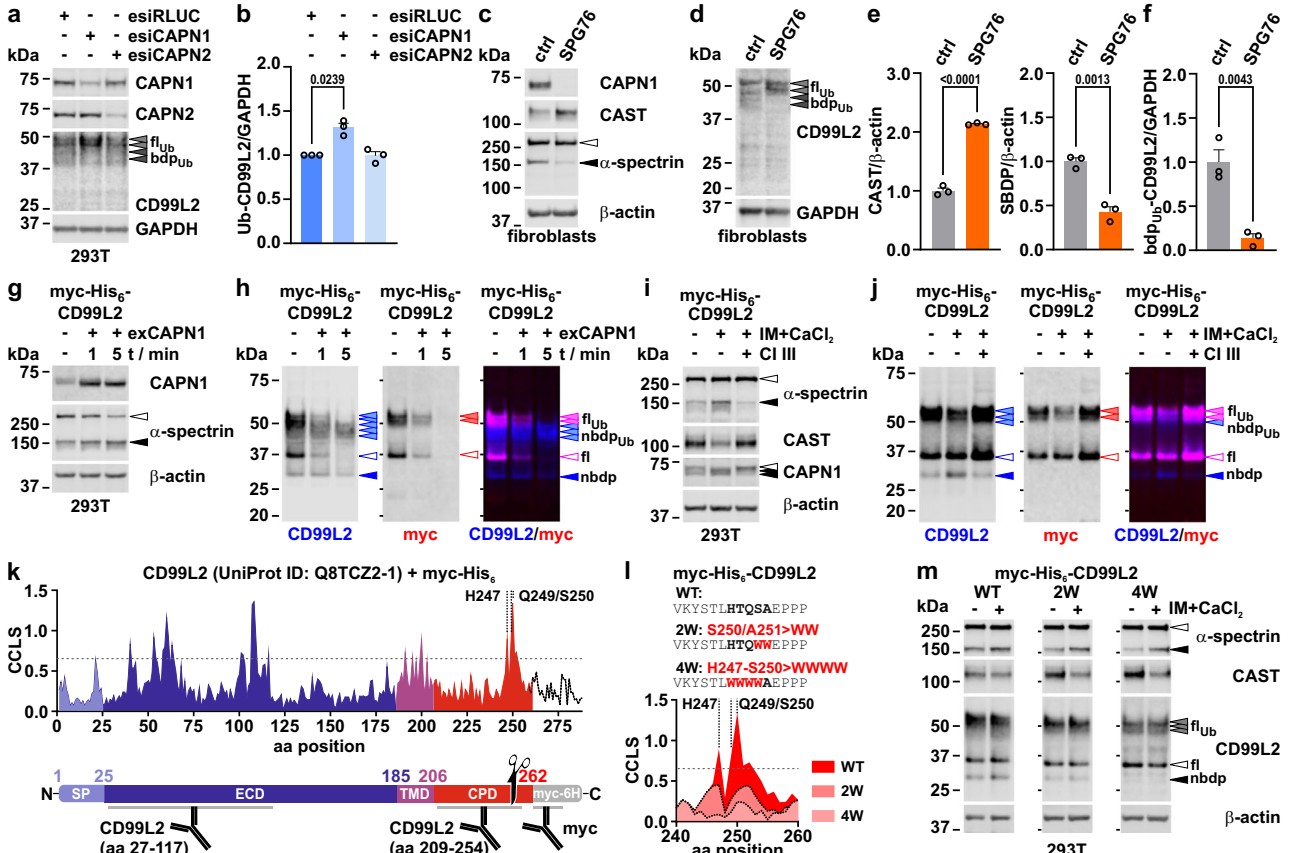

**Fig. 6 | CD99L2 is a CAPN1 substrate and cleaved at its cytoplasmic C-terminus.**
**a** Western blot of CAPN1, CAPN2 and CD99L2 in 293T cells transfected with esiRNAs against *CAPN1* (esiCAPN1), *CAPN2* (esiCAPN2) or *Renilla* luciferase (esiRLUC, control). GAPDH served as loading control. Grey arrowheads indicate ubiquitinated full-length (fl_Ub) CD99L2, and dark grey arrowheads cleaved ubiquitinated (bdp_Ub) CD99L2.
**b** Quantification of endogenous fl_Ub-CD99L2 upon CAPN1 or CAPN2 knockdown in 293T cells. n = 3 biologically independent experiments. One-sample *t*-test. **c, d** Western blots of CAPN1, CAST, and α-spectrin (**c**), and CD99L2 (**d**) in primary fibroblasts from a healthy control (ctrl) and a spastic paraplegia 76 (SPG76) patient. β-actin (**c**) or GAPDH (**d**) served as loading controls. White arrowhead indicates full-length (fl) α-spectrin and black arrowhead cleaved α-spectrin. **e, f** Quantification of CAST, calpain-cleaved α-spectrin (SBDP), and bdp_Ub-CD99L2. n = 3 passages. Student's *t*-test. **g, h** Western blots of CAPN1 and α-spectrin (**g**), and myc-His6-CD99L2 (**h**) from a time-dependent in vitro calpain cleavage assay in 293T cell extracts after exogenous CAPN1 (exCAPN1) administration. β-actin served as loading control. Light-blue/red/purple arrowheads indicate fl_Ub-CD99L2 or its ubiquitinated N-terminal breakdown products (nbdp_Ub); blue/purple/red-rimmed arrowheads indicate fl-CD99L2, and blue

arrowheads nbdp-CD99L2. **i, j** Western blots of α-spectrin, CAST, CAPN1 (**i**), and myc-His6-CD99L2 (**j**) in 293T cells after 1-h-ionomycin (IM)/CaCl2 treatment. Calpain inhibitor III (CI III) was used as a specificity control. β-actin served as loading control. **k, l** Computational prediction of calpain cleavage likelihood scores (CCLS) along the myc-His6-CD99L2 sequence (UniProt identifier: Q8TCZ2-1) (**k**) and for the C-terminal region (aa 240–260) containing double (2 W) or quadruple (4 W) tryptophan substitutions at predicted cleavage sites H247/Q249/S250 (**l**). Dotted line marks the prediction cut-off value (0.654). Schematic below in (**k**) shows myc-His6-CD99L2 aligned to the prediction graph. SP signal peptide, ECD extracellular domain, TMD transmembrane domain, CPD cytoplasmic domain. Black scissor indicates the predicted calpain cleavage site; antibody binding sites are highlighted. **m** Western blot of α-spectrin, CAST and CD99L2 in 293T cells expressing wild-type (WT), 2 W, or 4 W myc-His6-CD99L2 variants after 1-h-IM/CaCl2 treatment. β-actin served as loading control. White arrowheads indicate full-length α-spectrin and CD99L2; black arrowheads mark calpain-cleaved α-spectrin and nbdp-CD99L2. Bars represent mean ± s.e.m. All *t*-tests are two-tailed. P values are shown in the graphs. Source data are provided as a Source Data file.

protease CAPN1. Remarkably, these pathomechanistic links at the molecular level are perfectly paralleled by our clinical observations, suggesting a significant overlap between *CAPN1-* and *CD99L2-*associated phenotypes[16]. In contrast to previous studies[28], we demonstrate that the predominant PTM of CD99L2 is ubiquitination rather than glycosylation. Notably, CD99L2 is tightly regulated by both calpain-mediated cleavage and ubiquitination-dependent degradation via the proteasomal and autophagosomal pathways. Specifically, ubiquitination of CD99L2 appears to reduce its binding to CAPN1 and attenuate calpain system activation, consistent with established roles of this PTM in modulating protein-protein interactions[40]. This regulatory mechanism may help prevent excessive proteolytic activity, which has been implicated in neurodegenerative diseases such as polyglutamine disorders[32]. Previous studies had already identified two other membrane-bound proteins, Ttm50 in *Drosophila* and TRPC6, as calpain-activating proteins[41,42]. However, CD99L2 appears to specifically influence CAPN1. As no

channel-like properties or calcium homoeostasis-linked functions have yet been described for CD99L2, elucidating the precise mechanism by which CD99L2 activates calpains should be a key focus on future investigations. Furthermore, the complex regulatory mechanism involving the interplay of calpain-mediated cleavage, ubiquitination, and UPS-mediated turnover of CD99L2—summarised by our hypothetical schematic (Fig. 8k)—requires further dissection to identify and characterise its components. These may include the responsible DUBs, E3 ubiquitin ligases, and the exact sites of potentially modulatory PTMs in CD99L2. Interestingly, our initial data suggests a role for ataxin-3, the disease-associated protein in spinocerebellar ataxia type 3 and a calpain substrate itself[30], in regulating the ubiquitination status of CD99L2, pointing to a possible pathomechanistic overlap in neurodegeneration.

Our transcriptome analysis of patient-derived fibroblasts revealed a pronounced and specific dysregulation of genes, with strong enrichment for biological processes linked to neuronal function,

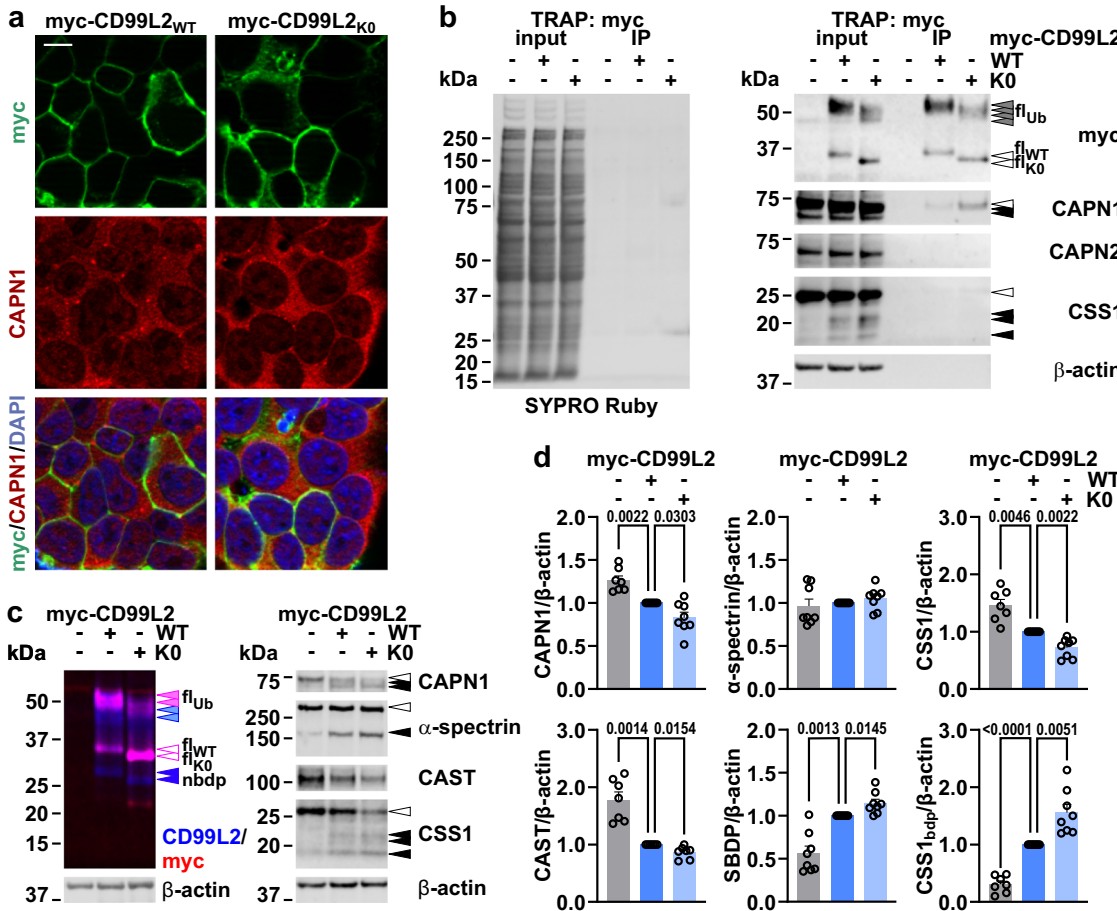

**Fig. 7 | CD99L2 ubiquitination modulates CAPN1 binding and calpain activation. a** Immunofluorescence microscopy of wild-type (WT) and lysine-free (K0) myc-His$_6$-CD99L2 and CAPN1 in 293T cells. CD99L2 (green) was detected via its myc tag, and CAPN1 (red) with a specific antibody. Nuclei (blue) were counterstained with DAPI. Scale bar, 10 μm. **b** Western blot of myc-TRAP-based immunoprecipitation (IP) of WT and K0 myc-His$_6$-CD99L2 from 293T cells. Total protein was visualised with SYPRO Ruby. Co-immunoprecipitated CAPN1, CAPN2 and CSS1 were analysed. β-actin served as loading control. Grey arrowheads indicate ubiquitinated full-length (fl$_{Ub}$) CD99L2, white arrowheads full-length proteins and black arrowheads their cleaved forms. fl$_{WT}$, full-length CD99L2$_{WT}$; fl$_{K0}$, full-length CD99L2$_{K0}$. **c** Western blot of CD99L2, CAPN1, α-spectrin, CAST and CSS1 in 293T cells expressing WT or K0 myc-His$_6$-CD99L2, or transfected with empty

vector. β-actin served as loading control. Light-purple arrowheads indicate fl$_{Ub}$-CD99L2, light-blue arrowheads cleaved N-terminal ubiquitinated forms (nbdp$_{Ub}$), purple-rimmed arrowheads WT and K0 full-length CD99L2 (fl$_{WT}$ and fl$_{K0}$), and blue arrowheads the N-terminal CD99L2 breakdown product (nbdp). White arrowheads indicate full-length proteins; black arrowheads indicate their cleaved forms. **d** Quantification of total CAPN1, full-length CAST, α-spectrin and CSS1, as well as calpain-cleaved α-spectrin (SBDP) and CSS1 (CSS1$_{bdp}$) in 293T cells expressing WT or K0 myc-His$_6$-CD99L2, or transfected with empty vector. $n = 7$ (empty vector: CAPN1, CAST, CSS1 and CSS1$_{bdp}$; WT CD99L2: CAST; K0 CD99L2: CAST) or $n = 8$ (all other conditions) biologically independent experiments. Bars represent mean ± s.e.m. Two-tailed one-sample $t$-test. $P$ values are shown in the graphs. Source data are provided as a Source Data file.

particularly the synapse. This transcriptional signature overlaps with synaptic pathways known to be affected by the loss of CAPN1, with loss-of-function variants in the encoding gene causing SPG76 and forms of cerebellar ataxia[16,32,36].

In conclusion, we establish loss or structural disturbances of CD99L2 due to hemizygous *CD99L2* loss-of-function variants as an additional factor in the CAPN1-linked pathway that manifests with neurological phenotypes. As dysregulated calpain function has been associated with common neurodegenerative disorders[43,44], a better understanding of calpain biology might also inform attempts to modulate calpain activity for treatment of, e.g. Alzheimer's and Parkinson's disease, where these proteases cause disease protein fragmentation and show pathology-related activation disturbances[17,18].

## Methods
### Cohorts
Patients with a clinical diagnosis of a rare MD who had received diagnostic testing between 10/2016 and 06/2022 at the Institute of Medical Genetics and Applied Genomics Tübingen were eligible for the study.

Only patients without a prior genetic diagnosis were investigated by ES and/or GS, with about 12% of probands having previously received gene panel testing at the Institute. Selection and assignment to subgroups was done according to the documented HPO terms (for details see Supplementary Table 1). The majority of cases studied by exome sequencing (ES) and/or genome sequencing (GS) were investigated under a multidisciplinary diagnostic care concept based on clinical expertise at university hospital centres for rare diseases[24]. Visualisation of phenotype space was done using uniform manifold approximation and projection (UMAP, Fig. 1a; for methods details see Supplementary information). Following identification of the novel disease gene, additional index cases were detected during routine diagnostics, extended search of in-house research datasets, and the RD-Connect Genome-Phenome Analysis Platform (GPAP)[10].

### Oversight
All procedures involving human subjects were approved by the ethics committee of the Medical Faculty of the University of Tübingen, Germany (Genome+, ClinicalTrial.gov #NCT04315727; 066/2021BO2 for

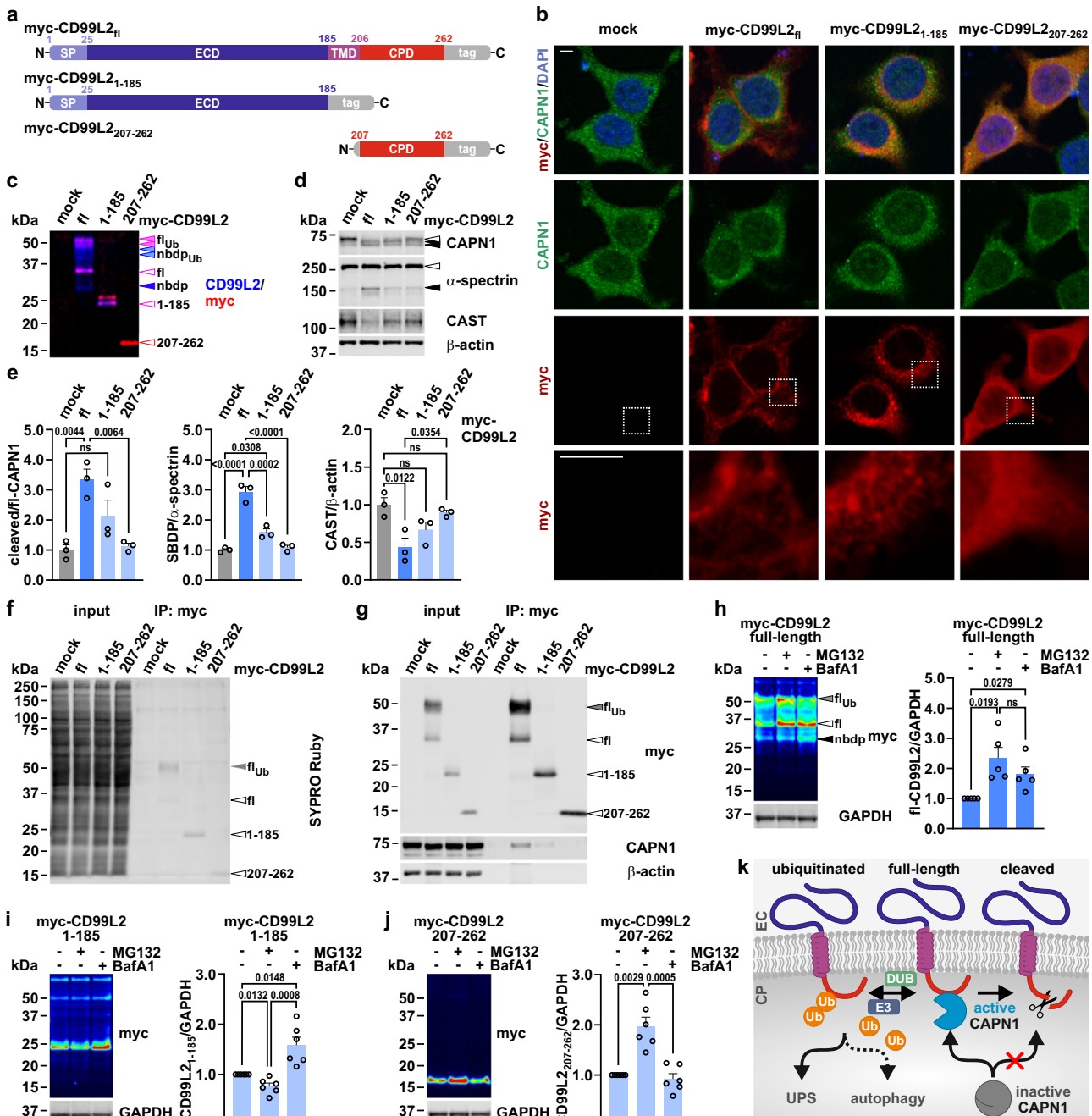

**Fig. 8 | CD99L2 truncation leads to aberrant cellular distribution, compromised calpain activation and reduced CAPN1 binding. a** Schematic of full-length (fl)- and truncated CD99L2 constructs, including N-terminal (CD99L2$_{1-185}$) or C-terminal (CD99L2$_{207-262}$) variants. SP signal peptide, ECD extracellular domain, TMD transmembrane domain, CPD cytoplasmic domain, tag myc-His$_6$ tag. **b** Immunofluorescence microscopy of fl- and truncated myc-His$_6$-CD99L2 together with CAPN1 in 293T cells. CD99L2 (red) was detected via its myc tag, and CAPN1 (green) using a specific antibody. Nuclei (blue) were counterstained with DAPI. Bottom row shows 5x magnifications of the white-boxed regions above. Scale bar, 5 µm. **c, d** Western blot of CD99L2 (**c**), and CAPN1, α-spectrin and CAST (**d**) in 293T cells expressing fl- or truncated myc-His$_6$-CD99L2. β-actin served as loading control. Light-purple arrowheads indicate ubiquitinated full-length (fl$_{Ub}$) CD99L2, light-blue arrowheads cleaved N-terminal ubiquitinated forms (nbdp$_{Ub}$), purple-rimmed arrowheads fl-CD99L2 or CD99L2$_{1-185}$, the red-rimmed arrowhead CD99L2$_{207-262}$, and the blue arrowhead the calpain-cleaved N-terminal CD99L2 breakdown product (nbdp). White arrowheads indicate full-length CAPN1 or α-spectrin; black arrowheads their cleaved forms. **e** Quantification of cleaved CAPN1, calpain-cleaved α-spectrin (SBDP), and CAST. $n = 3$ biologically independent

experiments. One-way ANOVA with Tukey's post hoc test. **f, g** Western blot of myc-TRAP-based co-immunoprecipitation (IP: myc) of fl- or truncated myc-His$_6$-CD99L2 and CAPN1 in 293T cells. Total protein was visualised using SYPRO Ruby. CD99L2 was detected via its myc tag. β-actin served as loading control. Grey arrowheads indicate fl$_{Ub}$-CD99L2, and white arrowheads fl- or truncated CD99L2. **h–j** Western blot and quantification of fl- and truncated myc-His$_6$-CD99L2 in 293T cells treated with MG132 or bafilomycin A1 (BafA1). CD99L2 was detected via its myc tag. GAPDH served as loading control. Black arrowhead indicates the N-terminal CD99L2 breakdown product (nbdp). $n = 5$ (fl-CD99L2) or $n = 6$ (CD99L2$_{1-185}$, CD99L2$_{207-262}$) biologically independent experiments. Two-tailed one-sample $t$- or Student's $t$-tests. **k** Model of CD99L2-CAPN1 interaction and CD99L2 proteostatic regulation. Inactive CAPN1 binds to unmodified fl-CD99L2 and becomes activated. Active CAPN1 cleaves CD99L2, removing its C-terminus, thereby limiting further CAPN1 activation. Ubiquitin ligases (E3) and deubiquitinases (DUB) balance ubiquitinated and unmodified CD99L2, ensuring protein degradation via the ubiquitin-proteasome system (UPS), or, to a lesser extent, autophagy, thus preventing excessive calpain activation. CP cytoplasm, EC extracellular space. Bars represent mean ± s.e.m. $P$ values are shown in the graphs; ns not significant. Source data are provided as a Source Data file.

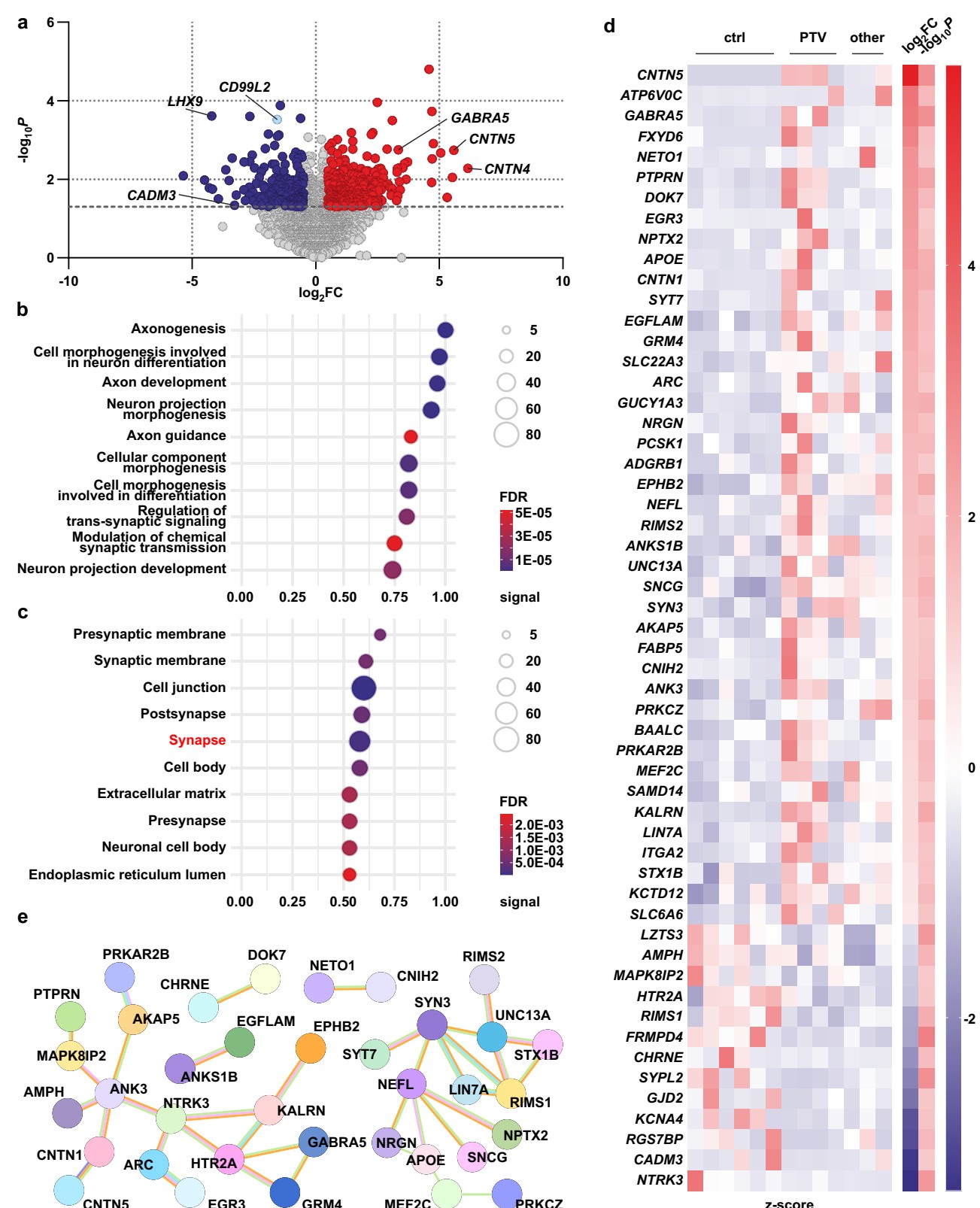

retrospective data analysis; 437/2018BO1 for Solve-RD and secondary use, ClinicalTrial.gov #NCT03491280; Biobank 199/2011 BO1, 423/2019BO1 for cell studies, Treat-HSP, ClinicalTrial.gov #NCT03981276). All probands or their legal representatives provided written informed consent for genetic testing. Consent for publication was obtained from all individuals whose detailed clinical data were presented. Participants did not receive compensation for their participation in the study. The

authors vouch for the accuracy and completeness of the data in this report.

## Visualisation of phenotypic space

As a first step, phenotypic characteristics annotated for known disease genes were downloaded from the HPO website (hp.obo data-version: hp/releases/2023-10-09 file: genes_to_phenotype.txt, downloaded on

**Fig. 9 | CD99L2 variants cause transcriptional disturbances in genes associated with neuronal function and the synapse. a** Transcriptomic analysis of fibroblasts derived from matched healthy controls and CD99L2 patients carrying protein-truncating variants, splice-site variants or deletions. Volcano plot displays $-\log_{10}P$ values versus expression $\log_2$ fold changes ($\log_2$FC) relative to controls. A total of 515 significant differentially expressed genes (DEGs) ($P \leq 0.05$; dotted horizontal line; $\log_2$FC $\leq -0.5$, $\log_2$FC $\geq 0.5$) are shown in colour (blue, downregulated; red, upregulated). *CD99L2* is highlighted in light blue. The top five DEGs related to neuronal function are additionally marked. **b**, **c** Functional enrichment analysis of DEGs from patient fibroblasts (protein-truncating variants, splice-site changes or deletions) compared to matched healthy controls using the STRING database (string-db.org). The top ten enriched biological processes (**b**) and cellular compartments (**c**) are shown as bubble plots. Signal values [defined as a weighted harmonic mean of the observed/expected ratio and $-\log_{10}$(FDR)], FDRs (bubble colour), and number of DEGs (bubble size) are indicated. **d** Heat map of synapse-related DEGs (gene ontology term) showing row-wise calculated *z*-scores, $\log_2$FC and $-\log_{10}P$ values for patient fibroblasts [protein-truncating variants (PTV), and splice-site changes or deletions (combined as "other")] versus matched healthy controls (ctrl). **e** STRING network of synapse-related DEGs in patients compared to matched healthy controls. Only DEGs with at least one interaction are shown. Light blue lines, curated database interactions; pink lines, experimentally determined interactions; orange lines, co-expression; green lines, text mining; lilac lines, protein homology. Network visualisations were obtained and adapted from STRING (string-db.org). Differential gene expression and statistical significance was calculated using EdgeR (see Supplementary Information for details). Source data are provided as a Source Data file.

16.11.2023). The data were merged with the annotations of individuals with ES or GS of the different subgroups (ataxia, dystonia, spasticity and the respective overlap groups) by treating each disease-ID as one individual. Subsequently, the similarities of the HPO terms between all pairs of individuals were calculated using the R package ontologySimilarity (version 2.5). The similarities were then converted into a distance matrix and projected to a four-dimensional space by uniform manifold approximation and projection (UMAP). Then, the first two dimensions of this projection were plotted using ggplot2 (https://github.com/GermanDemidov/phenotypes_plot).

### Details on genetic studies

**Conventional repeat expansion testing using FA.** Pseudonymized metadata from FA were retrieved from the local clinical information system (GenLab, Projodis) based on the documented respective analyses. These included diagnostically validated testing for the trinucleotide repeat expansion disorders spinocerebellar ataxia 1 (SCA1, *ATXN1*), spinocerebellar ataxia 2 (SCA2, *ATXN2*), Machado-Joseph disease (MJD/SCA3, *ATXN3*), spinocerebellar ataxia 6 (SCA6, *CACNA1A*), spinocerebellar ataxia 7 (SCA7, *ATXN7*), spinocerebellar ataxia 8 (SCA8, *ATXN8, ATXN8OS*), spinocerebellar ataxia 10 (SCA10, *ATXN10*), spinocerebellar ataxia 12 (SCA12, *PPP2R2B*), spinocerebellar ataxia 17 (SCA17, *TBP*), dentatorubral-pallidoluysian atrophy (DRPLA, *ATN1*), fragile X tremor/ataxia syndrome (FXTAS, *FMR1*) and Friedreich ataxia (FRDA, *FXN*). For the determination of the respective repeat length using fragment length analyses, the section of the respective gene (*ATN1, ATXN1, ATXN2, ATXN3, CACNA1A, ATXN7, TBP, ATXN8T/ATX8OS, ATXN10, PPP2R2B* and *FXN*) was amplified in a fluorescence-labelled PCR. The length of the respective fragments was determined by capillary electrophoresis (CEQ8000, Beckman Coulter) against an internal size standard (Size Standard 450 or 600, Beckman Coulter). For annotation of the respective alleles, the GenomeLab™ Genetic Analysis System software from Beckman Coulter. Singular signals are interpreted as homozygous genotypes. A heterozygous genomic deletion, which would mimic a homozygous presence, as well as an 'allelic drop-out' due to a very rare SNP among the primer binding sites, cannot be excluded with this method. The allele size is determined with an accuracy of ±1 repeat. For *ATXN8T/ATX8OS, ATXN10* and *FXN* fragment length analysis included also repeat primed PCR reactions. For amplification of the CGG repeat in *FMR1* the Asuragen AmplideX® PCR/CE FMR1 Kit was used, followed by capillary electrophoresis on the ABI Genetic Analyzer 3730xl with the ROX 1000 Size Ladder (Asuragen) and evaluation of the data in GeneMapper Software.

**Exome and genome analyses.** Exome and genome sequencing, as well as subsequent bioinformatic analysis were performed as described previously[25,45].

For ES, exonic and adjacent intronic regions were enriched from 3 μg genomic DNA using SureSelectXT Human All Exon v6 or v7 kits (Agilent Technologies, Santa Clara, California). Sequencing as 2 × 100 bp paired-end reads was performed on a HiSeq 2500 or NovaSeq 6000 system (Illumina, San Diego, California). Mean coverage of the target region was 132.81X.

For GS, 1 μg of genomic DNA was further processed using the TruSeq PCR-Free Library Prep kit (Illumina, San Diego, CA, USA) and the generated libraries were sequenced on a NovaSeq6000 system as 2 × 150 bp paired-end reads to an average 44.82X coverage. Deletions detected by exome sequencing in three individuals of family 2, 9 and 10 were subsequently confirmed by shallow GS with an average coverage of 5.27, 9.58 and 6.44 reads, respectively.

Mapping, variant calling and annotation of the data were performed using the megSAP pipeline (https://github.com/imgag/megSAP) developed at the Institute of Medical Genetics and Applied Genomics, University Hospital Tübingen, Germany. Details about the used tools and databases can be found in the megSAP documentation and have been published previously[25].

Clinical variant interpretation was conducted independently by two trained diagnostic molecular geneticists according to an in-house standard operating procedure. Variant classification was done according to the guidelines of the American College of Medical Genetics and Genomics (ACMG)[46,47]. For recessive disorders, the combination of a likely pathogenic or pathogenic variant with a 'hot VUS' on the other allele was considered a significant finding.

**Sanger sequencing.** Segregation analyses and carrier testing in CD99L2 families was by Sanger Sequencing. PCR primers were designed using the UCSC ExonPrimer-Primer Design Programme. Primer sequences and PCR conditions are provided below.

| Primer (exon) | Forward sequence | Reverse sequence | Position (hg38) |
|---|---|---|---|
| CD99L2-Ex1 | GCCTTA CCTCGCT GGACC | CAACCC AACCCAG GCCTC | chrX:150898515 + 150898911 |
| CD99L2-Ex6 | GGCAAG GATCCC ATGGGG | TGCCTG TGAGAAG CCAGC | chrX:150794789-150795387 |
| CD99L2-Ex7a | TGGGCA ACAT TAGCA CAAGT | GCATAATCA CCTTTTT CACCAGA | chrX:150793633-150793832 |
| CD99L2-Ex7b | TTTCT CCAGG TTTTTCAG ACAAGG | CAACTG GCAGGC TGGGAT | chrX:150793566-150793765 |
| CD99L2-Ex9 | CGTGCTT TGCAAA CCCCC | CCCTTC TTCACG CAGGCA | chrX:150775715 + 150776308 |

PCR conditions: 12.8 μl $H_2O$, 2.5 μl 10x PCR buffer + $MgCl_2$, 0.5 μl dNTP mix, 2.0 μl betaine (5 M), 2.0 μl Q-Solution (Qiagen), 0.2 μl Taq Polymerase (Thermo Fisher Scientific), 150 ng DNA sample, 1.0 μl each of 10 μM forward and reverse primer. PCR cycles: one cycle of 95 °C for 5 min, 10 cycles with 94 °C for 30 s, 65 °C (−1 °C per cycle) for 30 s, and 72 °C for 30 s, followed by 30 cycles with 94 °C for 30 s, 55 °C for 30 s, and 72 °C for 30 s, followed by 72 °C for 7 min and cooling to 10 °C.

**RNA sequencing and transcriptome analyses.** RNA was extracted from PAXgene tubes (14 patients and 54 male controls) and patients' ($n = 8$) and male control ($n = 6$) fibroblasts using the QIAsymphony RNA kits on a QIAsymphony SP with the protocol RNA CT 400 V7. To reduce preanalytical artefacts, cell lines were cultured and harvested according to an in-house standard operating procedure (also see section cell culture below). RNA quality was assessed with the Agilent 2100 Fragment Analyzer total RNA kit (Agilent Technologies, Inc., Santa Clara, United States). From 100 ng of total RNA, mRNAs were enriched using polyA capture on a NEBNext Poly(A) mRNA Magnetic Isolation Module (NEB). Libraries were prepared with Next Ultra II Directional RNA Library Prep Kits for Illumina (NEB) on a Biomek i7 (Beckman Sequencing). 215 pM cDNA libraries were sequenced on an Illumina NovaSeq6000 system as 2 × 100bp paired-end reads with ~50 million clusters per sample.

Bioinformatic processing and data interpretation were essentially done as previously described[25]. Generated RNA sequences were analysed with respect to aberrant expression, aberrant splicing and allelic imbalance using the megSAP pipeline (version 2022_08, https://github.com/imgag/megSAP). In brief, the ngs-bits tool collection (version 2022_08-92, https://github.com/imgag/ngs-bits) was used for quality control (ReadQC) and pre-processing (SeqPurge) of fastq files. STAR (version 2.7.10a, https://www.ncbi.nlm.nih.gov/pubmed/23104886, https://github.com/alexdobin/STAR/) was used for read alignment and detection of splice junctions, which were postprocessed with SplicingToBed. After mapping, MappingQC was used for quality control and Subread (version 2.0.3, https://pubmed.ncbi.nlm.nih.gov/30783653/, https://sourceforge.net/projects/subread/) for read counting based on an Ensembl gene annotation file (GRCh38, release 107, http://www.ensembl.org/index.html). Upon normalisation (megSAP) and quality assessment (RnaQC), expression values of genes and exons were compared with an in-house cohort (same tissue and processing system) using NGSDAnnotateRNA. For differential gene expression analysis, gene names were annotated and expression values (TPM, CPM, FPKM) filtered and normalised using EdgeR. Sample correlation is computed in order to detect outliers. Then, differential gene expression and statistical significance are calculated using EdgeR (https://www.rdocumentation.org/packages/edgeR/versions/3.14.0/topics/glmQLFit, https://rdrr.io/bioc/edgeR/man/glmQLFTest.html). Gene network and functional enrichment analysis of significantly differentially expressed genes ($P \le 0.05$; dotted horizontal line; $\log_2FC \le -0.5$, $\log_2FC \ge 0.5$) were performed using the Search Tool for the Retrieval of Interacting Genes/Proteins (STRING) database[35]. Data obtained from STRING-based analysis was graphically visualised by bubble plots that were generated using the ggplot2 package in R 4.2.2[48,49]. Full script can be provided upon request.

**Predictors of diagnostic yield.** About 1690 cases with ES and significant ($n = 394$) or non-significant findings ($n = 1296$) were analysed by using the least absolute shrinkage and selection operator (LASSO). Patients with uncertain candidate variants or genes were excluded from the analysis. The dataset was randomly divided into a training set incorporating 1368 cases (1053 unsolved, 315 solved) and a test set incorporating 322 cases (243 unsolved and 79 solved). The binary response of a case being solved or not solved was regressed on 17 subcategories of clinical features and controlling for gender (HPO mean, HPO count, ataxia, spasticity, dystonia, ataxia/dystonia, ataxia/

spasticity, dystonia/spasticity, ataxia/dystonia/spasticity, gender, 0–5 y, 5–20 y, 20–30 y, 30–40 y, 40–50 y, 50–60 y and 60–100 y). The HPO mean was calculated as the mean value of relative heights (derived from the relative position of HPO terms in the respective branch of the HPO tree) of all HPO terms assigned to one patient.

The model was fitted on the training data, and the penalty parameter was tuned via tenfold cross-validation. The resulting model was then applied to the independent test set and validated its predictive performance using the receiver operator characteristics (ROC) curve [AUC ROC 0.7 (0.64–0.761)]. We then refitted the model on the whole cohort of 1690 cases and validated the resulting model on an independent genome cohort of 261 cases (69 solved and 192 unsolved) [AUC ROC 0.609 (0.532–0.687)].

## Gene-burden analyses

**Data preparation.** MD individuals and controls were selected based on phenotypic criteria and diagnostic-grade quality (https://github.com/imgag/ngs-bits/blob/master/doc/GSvar/sample_quality_control.md) of sequencing datasets. Datasets from relatives and duplicate experiments were excluded from both groups. Variant calling and annotation were conducted uniformly using the megSAP pipeline as described above. High-impact variants included PTVs and canonical splice-site alternations, as well as homozygous exon deletions. Predicted deleterious variants were defined as missense variants having a CADD score ≥20. As different kits were used for enrichment, we restricted the analysis to genomic regions that were well represented across assays. We did not preselect for European ancestry.

**Gene-wise burden analysis.** The curated dataset comprised 2287 MD cases and 10,845 controls. To account for clinical heterogeneity among cases sharing the same genetic aetiology as observed with established MD entities, we merged the different MD subgroups into one with the aim of boosting power to identify statistical genetic associations. Association analyses were carried out for each gene separately, with pre-filtering of included variants adapted to represent recessive or dominant models of inheritance as well as different effect sizes of detected variation. Under a dominant model, variants with a likely minor allele frequency (MAF) <0.1 were included, with high-impact (PTVs and canonical splice sites) variants in the stringent and high-impact plus predicted deleterious (CADD score >20) variants in the relaxed analysis. Under a recessive model, only homozygous or hemizygous calls were included, predicting variants with a likely MAF <1%, again with high impact of combined effects for the stringent or relaxed analyses, respectively. In a fifth approach, we added high-quality CNVs for samples with a reference cohort correlation ≥0.9 for ES and ≥0.55 for GS, that were called with a log-likelihood of 15 or more and a copy-number status = 0. Alterations in CNV polymorphism regions or a putative MAF >1% as estimated from overlapping in-house deletions were excluded. To reduce the noise of false-positive prioritisation due to technical artefacts, CNVs were only considered as additional evidence for genes that were concomitantly impacted by small variants in at least two cases. Finally, case-control gene-wise burden analysis was performed on 2287 selected diagnostic cases and 10,845 controls using Fisher's exact test. To adjust for multiple testing, the P value was adjusted according to Bonferroni ($\alpha_{corr} = 1 - (1 - \alpha)^{1/n} \approx \alpha/n$) were $\alpha$ is the desired overall alpha level and $n$ is the number of hypotheses). When correcting for 10,165 genes affected by variants in the MD cohort, a P value $<4.9 \times 10^{-6}$ was considered as genome-wide significant. All P values were two-sided.

## Functional analyses methods

**Expression constructs and siRNAs.** For overexpression of human CD99L2 (canonical isoform 1, 262 aa; UniProt identifier: Q8TCZ2-1), its lysine-free variant (CD99L2 K0) in which all 14 lysine residues were mutated to arginine or its truncated forms spanning from amino acids

1–185 or 207–262, pcDNA3.1 vectors carrying cDNA for C-terminally myc-His$_6$-tagged forms of the proteins were generated (BioCat, Heidelberg, Germany). A C-terminally myc-DDK-tagged CD99L2 construct carried by a pCMV6-Entry vector was additionally purchased (RC207079, OriGene). Ataxin-3 overexpression was performed using pcDNA3.1 vectors harbouring C-terminally V5-tagged constructs with 15 or 77 glutamines[50]. Overexpression of calpastatin (CAST) was achieved using a pRK5 vector carrying the cDNA for full-length human CAST[51]. FLAG-tagged ubiquitin was overexpressed using a pcDNA3.1 vector. Empty pcDNA3.1 and pRK5 vectors were used for control transfections.

For marking the endoplasmic reticulum in fluorescence microscopy analysis, a psDsRed-ER vector (632409, Takara Bio Inc.) was used.

Gene knockdown was achieved using endoribonuclease-prepared siRNAs (esiRNAs) directed against human *CAPN1* (EHU032581) and *CAPN2* (EHU025391), with esiRNA against *Renilla* luciferase (MISSION® esiRNA EHURLUC) (all MISSION® esiRNAs, Merck) as control.

## Cell culture

After receiving written informed consent from the probands, fibroblast cell lines were established from skin punches according to routine procedures. HEK 293T (293T) cells (ATCC: CRL-11268), SH-SY5Y cells (ATCC: CRL-2266) and primary fibroblasts derived from patients and healthy control subjects were cultured in Dulbecco's modified Eagle's medium (DMEM), high glucose, GlutaMAX (10566016) supplemented with 10% (v/v) foetal bovine serum (FBS), 1% (v/v) non-essential amino acids (MEM NEAA) and 1% (v/v) Antibiotic-Antimycotic (A/A) (all Gibco®, Thermo Fisher Scientific) using 75 cm$^2$ cell culture flasks (REF 658175, Cellstar®, Greiner Bio-One) in 5% CO$_2$ and at 37 °C. For RNA extraction, fibroblasts were harvested from a 145/20 mm cell culture dish (REF 639160, Cellstar®, Greiner Bio-One) at 80% confluence. For protein extraction, fibroblasts were cultured in 100/20 mm tissue culture dishes (REF 353003, Falcon®, Corning Inc.). In experiments using 293T and SH-SY5Y cells, 6-well tissue culture plates (REF 353046, Falcon®, Corning Inc.) or 100/20 mm tissue culture dishes were employed.

**Cell treatments.** Transfections of 293T and SH-SY5Y cells were conducted for 72 h using Turbofectin 8.0 (Origene) for overexpression vectors or with Attractene transfection reagent (Qiagen) for esiRNA-based knockdown experiments according to the manufacturers' protocols.

Cellular *O*-GlcNAc cycling was targeted by inhibiting the enzyme *O*-GlcNAc transferase (OGT) with 10 μM OSMI-1 (Sigma-Aldrich) or the *O*-GlcNAcase with 10 μM Thiamet-G (Sigma-Aldrich) for 24 h. Blockage of proteasomal and autophagosomal pathways was conducted using 2.5 μM proteasome inhibitor MG132 (Merck) and 0.05 μM vacuolar-type H$^+$-ATPase inhibitor bafilomycin A1 (Enzo Life Science) for 24 h. Dimethyl sulfoxide (DMSO) served as the vehicle control. Before administration, all compounds were prediluted in Opti-MEM® I Reduced Serum Media (Gibco®, Thermo Fisher Scientific).

Cell-based calpain activation assays were performed as previously described[34]. Briefly, to activate endogenous calpains in transiently transfected 293T cells, the cell medium was replaced by Opti-MEM® I Reduced Serum Media containing 1.0 or 1.5 μM calcium ionophore ionomycin (in DMSO) (407950, Merck Millipore) and 5 mM CaCl$_2$. Cells were incubated under standard culture conditions for 1 h. For inhibitor controls, cells were pre-treated with 20 μM calpain inhibitor III (CI III; carbobenzoxy-valinyl-phenylalaninal; in DMSO) (Merck) starting 1 h before ionomycin administration. DMSO served as the vehicle control.

## Immunostaining and fluorescence microscopy

For immunocytochemistry staining, 293T cells were seeded on 8-well or 12-well chamber slides (80841 or 81201, Ibidi), transfected with

respective constructs, and cultured for 72 h. Subsequently, cells were pre-fixed by supplementing the medium with 0.4% (w/v) paraformaldehyde (PFA) and incubating at 37 °C for 10 min. Then, medium was aspirated, cells were washed once with 1× DPBS (Gibco®, Thermo Fisher Scientific) and incubated in 4% (w/v) PFA in 1× DPBS for 15 min. Next, the fixative was removed, and cells were washed three times with 1× DPBS for 5 min. Afterwards, cells were incubated in blocking/permeabilisation buffer (10% (w/v) BSA, 0.5% (v/v) Triton X-100, 0.02% (w/v) NaN$_3$ in 1× DPBS) at room temperature for 1 h. Cells were probed with mouse anti-myc-tag antibody (1:500; clone 9B11, #2276, Cell Signalling Technology) and rabbit anti-CAPN1 antibody (1:200; 10538-1-AP, Proteintech) in antibody diluent (1% (w/v) BSA, 0.5% (v/v) Triton X-100, 0.02% (w/v) NaN$_3$ in 1× DPBS) at 4 °C overnight. The next day, cells were washed four times with 1× DPBS and incubated with goat anti-mouse Alexa Fluor 488 (1:500; A32723, Thermo Fisher Scientific), goat anti-mouse Alexa Fluor 594 (1:500, A32742, Thermo Fisher Scientific) and goat anti-rabbit Alexa Fluor 488 (1:500, A27034, Thermo Fisher Scientific) secondary antibody at room temperature for 1 h. After washing four times with 1× DPBS for 5 min, the media chamber was removed, cells mounted with VECTASHIELD® Antifade Mounting Medium with DAPI (H-1200, Vector Laboratories) using coverslips and sealed with a transparent nail polish.

Fluorescent images were taken at 200× and 630× magnifications on an Axioplan 2 imaging microscope equipped with an ApoTome, Plan-Neofluar 20×/0.50 objective, Plan-Neofluar 40×/0.75 and Plan-Neofluar 63×/1.4 Oil objectives, and an AxioCam MRm camera, using the AxioVision 4.3 imaging software (all Zeiss).

## Protein extraction

Extraction of proteins was performed as follows. 293T cells were dissociated by gentle pipetting and transferred to 2 mL tubes. Cell pellets were obtained by centrifugation at 500×*g* for 5 min, followed by aspiration of the supernatant. Pellets were washed once with cold 1× DPBS. SH-SY5Y cells and fibroblasts obtained from healthy control subjects and patients were harvested by scraping cells in 1× DPBS and subsequent pelleting at 500×*g* for 5 min, followed by aspiration of the supernatant. Cell lysis was conducted by resuspending the pellet in RIPA buffer (50 mM Tris, pH 7.5, 150 mM NaCl, 10% glycerol, 0.1% (w/v) SDS, 0.5% sodium deoxycholate and 1% (v/v) Triton X-100) containing cOmplete™ protease inhibitor cocktail and PhosSTOP™ phosphatase inhibitors (both Roche), and incubated on ice for 15 min (293T and SH-SY5Y cells) or 30 min (fibroblasts). Afterwards, samples were centrifuged for 15 min (293T and SH-SY5Y cells) or 30 min (fibroblasts) at 4 °C and 20,000×*g*, and supernatants were transferred to a fresh, pre-cooled reaction tube. Protein concentrations were measured spectrophotometrically in a microtiter plate using Bradford reagent (Bio-Rad Laboratories). Samples were stored at 80 °C until further analysis.

## Immunoprecipitation

FLAG- and myc-tagged proteins were immunoprecipitated using DYKDDDDK Fab-Trap (ffa-20) and myc-Trap agarose (yta-20) (both ChromoTek/Proteintech Germany, Planegg-Martinsried, Germany). Ubiquitinated proteins were purified using Ubiquitin-Trap agarose (uta-20, ChromoTek/Proteintech). All immunoprecipitation experiments were performed according to the manufacturer's protocol with the following modifications: Transfected and/or treated 293T cells were harvested by resuspension in culture medium (content of a 100/20 mm cell culture dish) and subsequent centrifugation for 5 min at 4 °C and 500×*g*. Cell pellets were washed once with 1× DPBS and centrifuged again. Afterwards, cell pellets were lysed in 150 μL Trap lysis buffer (10 mM Tris pH 7.5, 150 mM NaCl, 0.5 mM EDTA, 0.5% IGEPAL CA−630) containing EDTA-free cOmplete protease inhibitor cocktail and PhosSTOP phosphatase Inhibitor cocktail (Roche). After measuring protein concentrations using Bradford reagent, lysate concentrations were adjusted with Trap lysis buffer to 5 μg/μL and then

further diluted in Trap dilution buffer (10 mM Tris, pH 7.5, 150 mM NaCl, 0.5 mM EDTA) containing EDTA-free cOmplete protease inhibitor cocktail and PhosSTOP phosphatase Inhibitor cocktail to a final concentration of 2 µg/µL. For immunoprecipitation, 500 µg total protein were incubated with 12.5 µL of agarose bead slurry, rotating end-over-end at 4 °C for 1 h. Subsequently, beads were washed three times with Trap wash buffer (10 mM Tris/Cl, pH 7.5, 150 mM NaCl, 0.5 mM EDTA and 0.05% IGEPAL CA-630) containing EDTA-free cOmplete protease inhibitor cocktail and heat-denatured at 70 °C for 10 min in 90 µL of 4× LDS sample buffer (1 M Tris, pH 8.5, 50% (v/v) glycerol, 8% (w/v) LDS, 2 mM EDTA and 0.1% (w/v) Orange G) mixed with Trap dilution buffer in a ratio 1:1 and supplemented with 0.1 M dithiothreitol (DTT). Samples were subsequently analysed by western blotting.

### Calpain cleavage assays in vitro

In vitro calpain cleavage assays were performed as previously described[52]. Briefly, 25 µg protein amount of 293T cell protein extracts after lysis in calpain reaction buffer (CRB) (20 mM HEPES/KOH, pH 7.6, 10 mM KCl, 1.5 mM $MgCl_2$ and 1 mM DTT). Reactions were prepared by diluting protein extracts in CRB and incubating them with 2 mM $CaCl_2$ and 100 ng of purified calpain-1 (208712, Merck) in a total volume of 20 µL at room temperature. Reactions were terminated by addition of 4× LDS buffer (1 M Tris, pH 8.5, 50% (v/v) glycerol, 8% (w/v) LDS, 2 mM EDTA and 0.1% (w/v) Orange G) in a ratio 3:1 with 0.1 M DTT, and heat-denaturing at 70 °C for 10 min. All samples were further analysed via western blotting.

### Western blotting

For western blotting, desired amounts of diluted protein extracts were mixed with 4× LDS sample buffer in a ratio of 3:1 supplemented with 0.1 M DTT and heat-denatured for 10 min at 70 °C. Afterwards, protein samples were electrophoretically separated using Bis-Tris gels and MOPS electrophoresis buffer (50 mM MOPS, 50 mM Tris, pH 7.7, 0.1% (w/v) SDS and 1 mM EDTA). Proteins were transferred on Amersham™ Protran™ Premium 0.2 µm nitrocellulose (Cytiva) using Bicine/Bis-Tris transfer buffer (25 mM Bicine, 25 mM Bis-Tris, pH 7.2, 1 mM EDTA and 15% (v/v) methanol) and a TE22 Transfer Tank (Hoefer) at 80 V at 25 °C for 2 h.

After transfer, membranes were stained for total protein using Ponceau S staining solution (0.1% (w/v) Ponceau S, 5% (v/v) acetic acid) for 1 min, or SYPRO™ Ruby Protein Blot Stain (Thermo Fisher) according to the manufacturer's protocols. Fluorescent SYPRO Ruby signals were detected using the LI-COR ODYSSEY® FC system (LI-COR Biosciences). If stained with Ponceau S, membranes were destained briefly with 1× TBS (10 mM Tris, pH 7.5, 150 mM NaCl). Afterwards, membranes were blocked for 45 min in 5% (w/v) skim milk powder (Sigma-Aldrich) diluted in 1× TBS at room temperature, and probed overnight at 4 °C with primary antibodies diluted in 1× TBS-T (TBS with 0.1% (v/v) Tween 20). All applied primary antibodies and their specifications are listed in Supplementary Table 2. Thereafter, membranes were washed with 1× TBS-T and incubated at room temperature for 1 h with the respective secondary IRDye® antibodies, goat anti-mouse 680LT (P/N 926−68020), goat anti-mouse 800CW (P/N 926-32210), goat anti-rabbit 680RD (P/N 926-68071) and goat anti-rabbit 800CW (P/N 926-32211) (all 1:5,000; LI-COR Biosciences). After final washing with 1× TBS-T, fluorescence signals were acquired using the LI-COR ODYSSEY®FC and quantified with Image Studio 4.0 software (both LI-COR Biosciences).

### In silico calpain cleavage site prediction

Computational analyses of features within the sequence of CD99L2 were based on its canonical isoform 1 (UniProt identifier: Q8TCZ2-1). For in silico calpain cleavage site prediction, Group-based Prediction System−Calpain Cleavage Detector (GPS-CCD) (available at http://ccd.biocuckoo.org/) or Labelling calpain substrate cleavage sites from

amino acid sequence using conditional random fields (LabCaS) (available at http://www.csbio.sjtu.edu.cn/bioinf/LabCaS/) tools were employed[33,53]. Likelihood scores for every amino acid position were graphically visualised using GraphPad Prism 9.1.2 (GraphPad Software).

### Statistics and reproducibility

For qualitative analyses, all experiments were independently repeated at least three times and yielded comparable results, ensuring validity and reproducibility. Quantitative data obtained from functional analyses were statistically analysed using GraphPad Prism 9.1.2 (GraphPad Software). Results are presented as individual data points with means and s.e.m. Statistical outliers indicated by the Grubbs test (α = 0.05) were excluded from the analysis. Two-tailed one-sample t-test, two-tailed Student's t-test or one-way ANOVA with Šídák's or Tukey's post hoc analysis were applied. Significance was assumed with a P value ≤0.05. Exact P values are shown in graphs. For further details, see the respective figure and figure legends, and the Source Data files.

### Reporting summary

Further information on research design is available in the Nature Portfolio Reporting Summary linked to this article.

## Data availability

The consented exome data from 259 cases and 178 in-house controls generated in this study have been deposited in the European Genome-Phenome Archive database under accession code EGAS00001003851. The exome data used in this study for matchmaking have been deposited in the European Genome-Phenome Archive database under accession code EGAC00001001319 [https://ega-archive.org/datasets/EGAD00001009769]. The exome data are available under restricted access for data privacy reasons; access can be obtained upon approval from the Solve-RD Data Access Committee (DAC) [https://solve-rd.eu/results/solve-rd-data/]. This requires completing and signing the Solve-RD Data Access Agreement (DAA) [https://solve-rd.eu/wp-content/uploads/2025/02/Solve-RD_DAA_250211_final.docx] and submitting it to the DAC office, as well as requesting access to the desired dataset through an EGA account. The review of data access requests can take up to 8 weeks. The DAA defines the conditions for accessing Solve-RD data, including sequence and genotype data, other omics data, phenotypic information, and pedigree data. It contains a research proposal, details about the applicant and any personnel who will access the data. The agreement must be signed by both the applicant and an authorised representative of their institution. The annex of the DAA describes the available datasets, any restrictions on use and the correct way to acknowledge use of the dataset. The consented genome data from 252 cases and eight in-house controls generated in this study have been deposited in the German Human Genome-Phenome Archive (GHGA) database under accession codes GHGAD30455110557096 and GHGAD63728375628676. The genome data were available under restricted access for data privacy reasons; access can be obtained by the data access committee (DAC). Requests for access can be sent via the GHGA portal [https://data.ghga.de/]. Access will be restricted to requests from authenticated users and assessed by the DAC, the local institutional review board, and the local data security office. The estimated timescale for data access is 3 months. The RNA CD99L2 expression read count data generated in this study are provided in the Source Data file. The RNA sequencing raw data are available under restricted access for privacy reasons, access can be obtained by researchers on request, subject to a confidentiality and data transfer agreement. Requests for access can be sent via email to the corresponding (tobias.haack@med.uni-tuebingen), with a 3-week timeframe for response. Unless objected to by the probands, identified clinically relevant variants were submitted to ClinVar [https://www.ncbi.nlm.nih.gov/clinvar/submitters/506385/]. The processed HPO

term data were available at the HPO GitHub repository [https://github.com/obophenotype/human-phenotype-ontology/releases/download/v2023-10-09/genes_to_phenotype.txt]. The metadata retrieved from the local clinical information system included information on age, test, disease group and diagnostic outcome and is provided in the paper and the Source Data. All additional data required to support the findings of this study are provided in the paper, the supplementary information, and the Source Data. Additional information can be provided upon request. Source data are provided with this paper.

## Code availability

Standard codes and custom scripts were used in this study and will be made available by the authors upon request. The NGS platform megSAP and ngs-bits are available on GitHub (https://github.com/imgag/megSAP and https://github.com/imgag/ngs-bits).

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

## Acknowledgements

The authors express their gratitude to all individuals participating in this study, their relatives, and caregivers. S.O., H.P.N., J.J.W., H.H. and T.B.H. received funding from the German Research Foundation (DFG; research grant numbers OS 647/1-1 to S.O., NG 101/6-1|WE 6585/1-1 to H.P.N. and J.J.W., HE8803/1–1 to H.H. and 418081722, 433158657, EJP-RD Artemis: 542553983 to T.B.H.). J.P. was supported by the Else Kröner-Fresenius-Stiftung Clinician Scientist programme "PRECISE.net." The funders had no role in the study design, data collection, data analysis, data interpretation or writing of the report. This study makes use of data and tools shared/provided through the RD-Connect GPAP, which received funding originally from the European Union Seventh Framework Programme (FP7/2007-2013) under grant agreement No. 305444. R.S. received funding from Bundesministerium für Bildung und Forschung (BMBF) through funding for the TreatHSP network (grant 01GM2209A), the European Health and Digital Executive Agency (HADEA) through funding for the European Rare Disease Research Alliance (ERDERA) (grant agreement 101156595), the National Institute of Neurological Disorders and Stroke (NINDS) and the National Institutes of Health (NIH) under Award Number R01NS072248, and the Clinician Scientist Programme PRECISE.net funded by the Else Kröner-Fresenius-Stiftung. The research of S.Z. reported in this publication was supported by the NINDS of the NIH under grant number [1U54NS148312], as well as the Spastic Paraplegia Foundation. T.B.H. received funding from the European Commission (Recon4IMD—GAP-101080997). The implementation of genome sequencing was supported in part by Illumina, Inc. through the Genome First approach and the Ge-Med concept. This study contributed to the Solve-RD project that has received funding from the European Union's Horizon 2020 research and innovation programme under grant agreement No 779257. Several authors of this publication are members of the European Reference Network for Rare Neurological Diseases (ERN-RND) - Project ID No 101085584 and/or of the European Reference Network on Rare for Rare Malformation Syndromes, Intellectual and Other Neurodevelopmental Disorders (ERN-ITHACA). ERN-ITHACA is funded by the EU4Health programme, grant agreement no. 101085231.

## Author contributions

J.J.W. and T.B.H. conceived the study. J.J.W. and T.B.H. jointly supervised research and data analyses. A.N., A.B., I.J.D., M.R., M.Rö, V.W., N.O., S.V., T.W.R., Z.K., C.H., M.K., N.W., L.Z., K.B., H.H., A.T., F.B., A.Be, A.Bi., F.B., R.B., D.B., N.C., C.C., I.C., K.C., M.D., N.E., M.E., R.J.F., T.F., Z.F., D.G., U.G., K.G., A.G., M.H., S.H., U.H., Y.H., W.H., B.H., R.H., V.H., D.J., N.K., C.K., M.K., J.K., C.K., K.K., H.K., U.K., P.K., M.K., H.K., A.K., L.L., A.L., N.L., T.L., J.M., E.M., Ev.M., I.M., K.M., D.M., S.M., B.O., M.G.P., M.Ph., F.R., M.R., A.R., C.S., B.S., A.S., E.M.C.S., V.S., S.S., K.S., C.S., A.T., S.W., C.W., S.Z., S.Zi, R.A.H., M.D., F.D., A.D., H.G., B.H., H.J., T.K., T.Klop., X.K., G.K., A.K., I.K., H.N., G.W., K.E.Z., S.K., M.A., M.B., J.W., D.T., M.S., B.W., R.S., L.J.S. and O.R. collected patients' materials and investigated phenotypes. F.H., N.C., B.Mac and A.V. coordinated sequencing and proteome experiments. B.M., C.D., N.D., A.H., K.R., F.H., I.B., S.B., R.B., R.J.F., L.G., S.G., K.G., M.R., S.Z., O.R. and T.B.H. performed clinical variant interpretation. B.M., G.D., M.S., J.A., H.K., P.K. and T.B.H. performed statistical genetic data analyses. G.D., M.S., L.S., C.G., J.A. and S.O. developed the bioinformatics and computational pipelines. R.D.I., C.H., F.F., S.P., P.P., M.G.P. and J.J.W. performed cellular studies. B.M., R.D.I., J.J.W. and T.B.H. drafted the manuscript. All authors reviewed and approved the paper.

## Funding

## Competing interests

S.K. has received lectures honoraria from Esteve Pharma, Ipsen, AbbVie, Roche, Stadapharm and Bial. M.S. has received consultancy honoraria from Ionis, Prevail, Orphazyme, Servier, Reata, GenOrph and AviadoBio. S.O. received reimbursement for travel expenses and payment for conference presentations from Illumina, Inc. and Oxford Nanopore Technologies. All declared honoraria are unrelated to the present manuscript. The remaining authors declare no competing interests.

## Additional information

Benita Menden[1,68], Rana D. Incebacak Eltemur[1,2,68], German Demidov[1], Marc Sturm[1], Joohyun Park[1], Chrisovalantou Huridou[1,2], Florian Fath[1,2,3], Astrid Nümann[4], Alexander Baumann[5], Illja J. Diets[6], Claudia Dufke[1], Martin Regensburger[7], Maria Rönnefarth[4], Vera Wilke[8], Nienke van Os[9], Stefan Vielhaber[10], Tim W. Rattay[5,11,12], Zacharias Kohl[13], Susana Peralta[1], Priscila Pereira Sena[1], Melanie Kellner[12], Nadine Weissert[12], Andreas Traschütz[14], Lena Zeltner[12], Kai Boelmans[15,16], Natalie Deininger[1], Leon Schütz[1], Caspar Gross[1], Ana Beatriz Hinojosa Amaya[1], Katrin Raupach[1], Holger Hengel[12], Florian Harmuth[1], Jakob Admard[1], Ingrid Bader[1], Sarah Baumann[1], Friedemann Bender[12], Andrea Bevot[17], Almut Bischoff[18], Felix Boschann[19], Rebecca Buchert[1], Daniel Buchzik[20], Nicolas Casadei[1], Claudia B. Catarino[18], Isabell Cordts[21], Kirsten Cremer[22], Marion Doebler-Neumann[17], Nadja Ehmke[23], Miriam Elbracht[24], Ruth J. Falb[1], Thomas Feindt[25], Zofia Fleszar[12], Lea Gerstner[1], Dieter Gläser[26], Ute Grasshoff[1,27], Sarah Grosch[1], Kathrin Grundmann[1], Alexander Gutschalk[28], Manja Haaga[17], Stefanie Hayer[12], Ute Hehr[29], Yorck Hellenbroich[30], Wolfram Henn[31], Barbara Herr[32], Rebecca Herzog[33], Veronka Horber[17], Jonas Deppe[21], Nadja Kaiser[17], Christiane Kehrer[17], Martin Kehrer[1], Jan Kern[17], Christoph Keßler[12], Katharina Khuller[34], Hannah Klinkhammer[35], Urania Kotzaeridou[36], Peter Krawitz[35], Martina Kreiss[22], Hanna Küpper[17], Alice Kuster[37], Lucia Laugwitz[17], Anne Lesemann[4], Nadine Lichey[38], Tobias Linden[39], Boris Macek[40], Janine Magg[17], Elisabeth Mangold[22], Eva Manka[41], Iris Marquardt[39], Karl Mehnert[26], David Mengel[12], Susanne Morlot[42], Barbara Oehl-Jaschkowitz[43], Martje G. Pauly[44], Melanie Philipp[45], Florentine Radelfahr[18], Maren Rautenberg[1], Angelika Riess[46], Carsten Saft[3], Beate Schlotter-Weigel[18], Axel Schmidt[22], Eva M. C. Schwaibold[47], Veronika Spahlinger[12], Stephanie Spranger[38], Katharina Marie Steiner[48], Claudia Stendel[18], Andreas Thieme[49], Andreas Tzschach[32], Ana Velic[40], Sarah Wiethoff[50], Carlo Wilke[14], Stephan Züchner[51], Simone Zittel[52], Solve-RD consortium*, Ralf A. Husain[53], Marcus Deschauer[21], Felix Distelmaier[54], Andreas Dufke[1], Holm Graessner[1,27], Bernhard Hemmer[21], Heike Jacobi[28], Thomas Klockgether[55], Thomas Klopstock[18,56,57], Xenia Kobeleva[55,58], Georg-Christoph Korenke[39], Alma Kuechler[34], Gregor Kuhlenbäumer[5], Ingo Kurth[24], Huu Phuc Nguyen[2], Gilbert Wunderlich[59], Kirsten E. Zeuner[5], Stephan Klebe[48,60], Michaela Auer-Grumbach[61], Michaela Butryn[62], Jürgen Winkler[7,63], Dagmar Timmann[49], Matthis Synofzik[14,64], Bart van de Warrenburg[9], Rebecca Schüle[65,66], Ludger Schöls[12,64], Stephan Ossowski[1], Olaf Riess[1,27,67], Jonasz J. Weber[1,2,69] ✉ & Tobias B. Haack[1,27,67,69] ✉

[1]Institute of Medical Genetics and Applied Genomics, University of Tübingen, Tübingen, Germany. [2]Department of Human Genetics, Ruhr University Bochum, Bochum, Germany. [3]Department of Neurology, St. Josef Hospital, Ruhr University Bochum, Bochum, Germany. [4]Department of Neurology, Charité - Universitätsmedizin Berlin, corporate member of Freie Universität Berlin and Humboldt-Universität zu Berlin, Berlin, Germany. [5]Department of Neurology, University Hospital Schleswig-Holstein, Kiel, Germany. [6]Department of Human Genetics, Radboud University Medical Center, Nijmegen, Netherlands. [7]Department of Molecular Neurology, University Hospital Erlangen, Erlangen, Germany. [8]Department of Neurology and Center for rare diseases, University Hospital Tübingen, Tübingen, Germany. [9]Department of Neurology, Radboud University medical center, Nijmegen, Netherlands. [10]Department of Neurology, University Hospital Magdeburg, Magdeburg, Germany. [11]Center for Rare Diseases, University Hospital Kiel, Kiel, Germany. [12]Department of Neurology, University Hospital Tübingen, Tübingen, Germany. [13]Department of Neurology, University Hospital Regensburg, Regensburg, Germany. [14]Division Translational Genomics of Neurodegenerative Diseases, Hertie Institute for Clinical Brain Research and Center of Neurology, University Hospital Tübingen, Tübingen, Germany. [15]Department of Neurology, University Hospital Würzburg, Würzburg, Germany. [16]Department of Neurology, Klinikum Bremerhaven-Reinkenheide, Bremerhaven, Germany. [17]Department of Neuropediatrics, General Pediatrics, Endocrinology, Diabetology, Social Pediatrics, University Children's Hospital Tübingen, Tübingen, Germany. [18]Friedrich-Baur-Institute at the Department of Neurology, LMU University Hospital, Ludwig-Maximilian University (LMU) Munich, Munich, Germany. [19]Institute of Human Genetics, Charité - Universitätsmedizin Berlin, corporate member of Freie Universität Berlin and Humboldt-Universität zu Berlin, Berlin, Germany. [20]Sozialpädiatrisches Zentrum, Diakoneo Diak Klinikum Schwäbisch Hall, Schwäbisch Hall, Germany. [21]Department of Neurology, Technical University Munich, Munich, Germany. [22]Institute of Human Genetics, University of Bonn, Medical Faculty & University

Hospital Bonn, Bonn, Germany. [23]Institute for Medical Genetics and Human Genetics, Charité - Universitätsmedizin Berlin, corporate member of Freie Universität Berlin and Humboldt-Universität zu Berlin, Berlin, Germany. [24]Center for Human Genetics and Genomic Medicine, Uniklinik RWTH Aachen, Aachen, Germany. [25]Praxis Neurologie Magdeburg, Magdeburg, Germany. [26]genetikum, Neu-Ulm, Germany. [27]Center for Rare Diseases, University of Tübingen, Tübingen, Germany. [28]Department of Neurology, University Hospital Heidelberg, Heidelberg, Germany. [29]Zentrum für Humangenetik Regensburg, Regensburg, Germany. [30]Department of Human Genetics, University Hospital Schleswig-Holstein, Lübeck, Germany. [31]Institute of Immunology and Genetics, Practice for human genetics in the Medicushaus Kaiserslautern, Kaiserslautern, Germany. [32]Institute of Human Genetics, University Hospital Freiburg, Freiburg, Germany. [33]Department of Neurology, University Hospital Lübeck, Lübeck, Germany. [34]Institute of Human Genetics, University Hospital Essen, Essen, Germany. [35]Institute for Genomic Statistics and Bioinformatics, University of Bonn, Medical Faculty & University Hospital Bonn, Bonn, Germany. [36]Department of Pediatrics, University of Heidelberg, Heidelberg, Germany. [37]Department of Pediatrics, University of Nantes, Nantes, France. [38]Praxis für Humangenetik, Bremen, Germany. [39]Department of Pediatrics, University of Oldenburg, Oldenburg, Germany. [40]Proteome Center Tübingen, University of Tübingen, Tübingen, Germany. [41]Department of Pediatrics, University Hospital Essen, Essen, Germany. [42]Institute of Human Genetics, University of Hannover, Hannover, Germany. [43]MVZ Labor Saar GmbH, Praxis für Humangenetik, Homburg, Germany. [44]Institute of Systems Motor Science, University of Lübeck, Lübeck, Germany. [45]Department of Experimental and Clinical Pharmacology and Pharmacogenomics, University of Tübingen, Tübingen, Germany. [46]Institute of Medical Genetics and Applied Genomics, University Hospital Tübingen, Tübingen, Germany. [47]Institute of Human Genetics, Heidelberg University, Heidelberg, Germany. [48]Department of Neurology, University Hospital Essen, Essen, Germany. [49]Department of Neurology and Center for Translational Neuro- and Behavioral Sciences (C-TNBS), University Hospital Essen, University of Duisburg-Essen, Essen, Germany. [50]Department of Neurology, University Hospital Münster, Münster, Germany. [51]Department of Human Genetics and John P. Hussman Institute for Human Genetics, University of Miami Miller School of Medicine, Miami, USA. [52]Department of Neurology, University Medical Center Hamburg-Eppendorf, Hamburg, Germany. [53]Department of Neuropediatrics, Jena University Hospital, Jena, Germany. [54]Department of General Pediatrics, Neonatology and Pediatric Cardiology, Medical Faculty and University Hospital Düsseldorf, Heinrich-Heine-University, Düsseldorf, Germany. [55]Department of Neurology, University of Bonn, Medical Faculty & University Hospital Bonn, Bonn, Germany. [56]German Centre for Neurodegenerative Diseases (DZNE) Site Munich, Munich, Germany. [57]Munich Cluster for Systems Neurology (SyNergy), Munich, Germany. [58]Computational Neurology Group, Medical Faculty, Ruhr University Bochum, Bochum, Germany. [59]Department of Neurology and Center for Rare Diseases, University of Cologne, Faculty of Medicine and University Hospital, Köln, Germany. [60]Department of Neurology, Knappschaftskrankenhaus Recklinghausen, Recklinghausen, Germany. [61]Department of Orthopedics and Trauma-Surgery, University Hospital Vienna, Vienna, Austria. [62]Deutsches Zentrum für Neurodegenerative Erkrankungen Magdeburg, Magdeburg, Germany. [63]Center for Rare Diseases Erlangen (ZSEER), University Hospital Erlangen, Erlangen, Germany. [64]German Center for Neurodegenerative Diseases (DZNE) Site Tübingen, Tübingen, Germany. [65]Hertie Institute for Clinical Brain Research and Center of Neurology, University Hospital Tübingen, Tübingen, Germany. [66]Division of Neurodegenerative Diseases and Movement Disorders, Department of Neurology, Heidelberg University Hospital, Heidelberg, Germany. [67]Genomics for Health in Africa (GHA), Africa-Europe Cluster of Research Excellence (CoRE), Tübingen, Germany. [68]These authors contributed equally: Benita Menden, Rana D. Incebacak Eltemur. [69]These authors jointly supervised this work: Jonasz J. Weber, Tobias B. Haack *A list of authors and their affiliations appears at the end of the paper.
✉e-mail: jonaszjeremiasz.weber@rub.de; tobias.haack@med.uni-tuebingen.de

## Solve-RD consortium

German Demidov ®[1], Marc Sturm ®[1], Joohyun Park[1], Nienke van Os[9], Melanie Kellner[13], Andreas Traschütz[14], Leon Schütz ®[1], Holger Hengel ®[13], Isabell Cordts[21], Rebecca Herzog[33], Martje G. Pauly ®[44], Carlo Wilke[14], Marcus Deschauer[21], Holm Graessner ®[1,27], Matthis Synofzik ®[14,67], Bart van de Warrenburg[9], Rebecca Schüle[64,65], Ludger Schöls[13,67], Stephan Ossowski ®[1], Olaf Riess ®[1,27,66] & Tobias B. Haack ®[1,27,66,69]✉

A full list of members and their affiliations appears in the Supplementary Information.

