## [Transparent Peer Review file · Nature Communications]

Loss-of-function variants in the CAPN1 activator CD99L2 cause X-linked spastic ataxia

Corresponding Author: Dr Tobias Haack

Version 0:

Reviewer comments:

Reviewer #1

(Remarks to the Author)

In this manuscript, Menden et al compared different genetic testing strategies in a very large cohort of 2811 subjects affected by various movement disorders, exploiting the efforts of a national genome sequencing program. The main outcomes of this study are:

- The overall diagnostic yield was nearly 20%, with the best outcome for spastic paraplegia and the worst for dystonia
- Genome sequencing significantly increased the diagnostic rate by about 7.5%, mainly due to structural variants and expansions not detectable with conventional WES
- Gene-burden analysis of negative cases identified loss of function variants in the CD99L2 gene as causative of a novel form of X-linked spastic ataxia
- Functional studies demonstrated the role of CD99L2-encoded protein as activating interactor of the protease CAPN1, already linked to forms of recessive spastic paraplegia and ataxia.

Comments to the authors

- 1) I find the general term “movement disorders” questionable in this study. The authors selected three major categories, which are dystonia, cerebellar ataxia and spastic paraplegia. While the inclusion of the latter is justified by the relevant overlap with cerebellar ataxia phenotype, spastic paraplegia cannot be strictly considered a “movement disorder”. On the other hand, other movement disorders were excluded from the HPO term selection, namely chorea, myoclonus and parkinsonism, just to name the most relevant ones from a genetic point of view. This should be clearly acknowledged and discussed / justified, as it should be made clear that the reported diagnostic yield cannot be generalized to rare movement disorders but only to a subset of them. I would suggest the title should also be changed to reflect this.
- 2) The authors mention that panel analysis, likely performed in the early days before the widespread diffusion of WES, were not counted. However, it is likely that a subset of patients underwent NGS-panel analysis from this cohort. Can the authors provide an estimate of these figures? Did all patients who previously tested negative at panel analysis undergo WES or WGS? Conversely, were patients diagnosed through panel-based analysis included or excluded by the present study? As this is a large nationwide-based project with likely “epidemiological” implications, it is important to have a precise knowledge of inclusion and exclusion criteria.
- 3) Supplementary Figure 2: does 2B panel refer to the age at onset? Please check the letters in the panels, as they do not always correspond to the legend.
- 4) Supplementary Figure 3: does 3A panel refer to the age at onset? 3B: why patients with no significant findings were tested later? Is there any correlation with the phenotype (HPO categories), or the age at onset? Also in panel 3C, explain whether this is age at onset or age at diagnosis. Same comments for figure 4.
- 5) In Supplementary Figures 2 and 4, please explain the criteria to assign “candidate genes” (yellow colour). How many of these candidate genes were validated / confirmed? They should be reported in a table.
- 6) Page 7, Gene-burden analysis refers to an extended prioritization list presented in Supplementary Table 3, but such Table is missing. Similarly, I cannot find Supplementary Table 4, quoted on page 7.
- 7) Page 7: Please check the number of index cases as they seem to be 21 (8+8+1+4), while later and in the figures the total number of families are 20.
- 8) Legend to Figure 2: please correct “blue arrowheads in (r) and (d)” to “blue arrowheads in (r) and (s)”
- 9) Non-truncating variants are of interest and should be characterized more in detail. Figure 2m and 2n show that RNA levels from blood or fibroblasts of patients carrying splicing variants or a multiexon deletion are significantly reduced compared to controls (although not abolished as shown for truncating variants). Western blotting (shown for three variants) lack the wild type band but show weaker, smaller bands. Only some variants were assessed at the transcriptional level

(Supplementary Figure 8). Were the bands observed on Western blotting correspond to the transcripts detected on RNAseq? Supplementary Figure 8 should be expanded to comprehensively describe the impact of non-truncating variants at the cDNA and protein level (by the way, on figure 2, panel m refers to c.535G>C while the legend refers to c.535G>A – please check)

10) Please comment why autophagy inhibition results in a reduction of ubiquitinated CD99L2 while the unmodified protein increases (Figure 3d).

11) Legend to figure 3: please move description of grey arrowheads/white arrowheads/black arrowheads to panels a-c.

12) Figure 4h-j: can the authors explain this divergent behavior of truncated CD99L2 in response to proteasome or autophagy inhibition?

13) Figure 4k: it would be useful to have a brief explanation of this schematic cartoon to help the readers draw the right conclusion about the postulated mechanism

14) Supplementary Figure 9: it seems that anti-CD99L2 antibody does not recognize any endogenous protein either in 293T cells or SH-SY5Y cells (mock transfected lane is perfectly clear not only in the myc-antibody panels but also in the CD99L2 panels and overexposure shows a very faint modified band). Is the protein expected to have a very rapid turnover in cells? Figure 3C seems to show that even inhibition of proteasome or autophagy do not result in increased levels of endogenous protein in mock-transfected cells. However, in Supplementary Figure 12 it seems that endogenous CD99L2 is detected in control and SPG76 fibroblasts. Please provide an explanation for these observations or clarify in the text

15) The authors discuss a cellular mislocalization of truncated forms of CD99L2 compared to wild type. By assessing colocalization with an ER-marker, it seems that truncated forms of the protein have increased co-localization with ER compared to wild type, however this is not explained in the text. Why the authors specifically assessed colocalization with ER? Have they also tested colocalization with CAPN1?

Reviewer #2

(Remarks to the Author)

This manuscript describes the approach to increase diagnostic power for rare diseases by rationally combining the measures to detect potential cause at the level of gene sequence and their association with clinical phenotypes. In their experimental settings using movement disorders as study subjects, improvements in diagnostic yields were suggested and the novel disease gene, CDL99L2, was identified. Then, properties of the gene product were explored demonstrating that CD99L2 is constitutively ubiquitinated and its turnover is regulated by proteasome and autophagy. In addition, CD99L2 interacts with CAPN1, a catalytic subunit of intracellular protease calpain-1, and activates it. Further, truncated mutants of CD99L2, either the N- or C-terminal, were shown to be defective in localization, calpain interaction/activation as well as ubiquitination. It is suggested that pathogenicity of the CDL99L2 variants identified in the study is manifested via dysregulation of calpain and/or compromise in degradative regulation of CDL99L2.

The first part of the study shows the impact of systematically performed evaluation of the huge sets of sequence data. Identification of a novel causative gene, CDL99L2, for the movement disorder is a great outcome without question. In the following experiment, basic properties of CDL99L2 was examined, however, the results as shown in Fig. 4 are not sufficient to support what the authors claim in discussion, i.e., functional investigation on CDL99L2 is missing. Therefore, it is not clear if there is any real association between the proteolytic systems, either proteasome, autophagy or calpain, and CD99L2. To demonstrate the relevance of the present result in terms of pathogenic mechanism stemming from the CDL99L2 variants, following points, at minimum, should be clarified; is there any functional relationship between ubiquitination and CAPN1-interaction of CDL99L2?, in terms of calpain-1 activation function of CDL99L2, what cellular functions are compromised in the cells from patients harboring CD99L2 mutations? (is activation of calpain-1 impaired ?), what is functional impact of ubiquitination of CD99L2 function?

Comments on minor points:

(line 176) The case number for FA should be 1,067 (as in Supl. Fig. 1).

(line 270) The change in molecular weight by ubiquitination is ca. 22 kDa at maximum corresponding to 3 ubiquitins, therefore, “highly” is unnecessary.

(line 226 in supplement) What mutation(s) does “lysine-free” stand for?

(line 445 in supplement) (l) should be (i).

In Figure legends, definitions for abbreviation and labelling (e.g., SBDP, “full”, “**” for statistical significance, etc.) should be provided when they first appeared.

Methods used for statistical significance should be consistently provided (missing in some figure legends).

What does labelling “**” in pedigree schemes in Fig. 1(a-k) and Fig. S7(a-i) indicate?

What is the rational in testing spectrin cleavage and including CAPN10 in the experiment shown in Fig. 3 etc.?

What does “mislocalization” (as in descriptions for Fig. 4b and Fig.13a) mean exactly?

The title could be modified to better represent the article's content.

Version 1:

Reviewer comments:

Reviewer #1

(Remarks to the Author)

I appreciate the work made by the authors to improve the manuscript and address the criticisms raised by the reviewers. A

lot of additional experimental work has been added and several issues have been clarified. I am overall satisfied of the data presented here.

Enza Maria Valente, MD, PhD

Reviewer #2

(Remarks to the Author)

The revised manuscript shows great improvement in the overall construction as well as data presentation. It is greatly appreciated that the authors have added much work and I believe that an important lead for the future studies is established.

A few minor corrections:

Page 6, line 165; the number "1,076" should be "1,067" (as in line 181).

Page 7, line 224; considering the expression "Of the 3 associations identified, 2... were already known...and 1... was ... unidentified (line 236-238), "three known" should be "two known"?

Page 9, line 331; should be "Fig. 3h, i"?

Changing the order of Supplementary Tables could better suit the flow of the manuscript; the present Table S2 showing a list of used antibodies could be Table S4 since the content is not referred to in the main text and stands as a component in supplementary methods.

Reviewer #1 (Remarks to the Author)

In this manuscript, Menden et al compared different genetic testing strategies in a very large cohort of 2811 subjects affected by various movement disorders, exploiting the efforts of a national genome sequencing program. The main outcomes of this study are:

- The overall diagnostic yield was nearly 20%, with the best outcome for spastic paraplegia and the worst for dystonia
- Genome sequencing significantly increased the diagnostic rate by about 7.5%, mainly due to structural variants and expansions not detectable with conventional WES
- Gene-burden analysis of negative cases identified loss of function variants in the CD99L2 gene as causative of a novel form of X-linked spastic ataxia
- Functional studies demonstrated the role of CD99L2-encoded protein as activating interactor of the protease CAPN1, already linked to forms of recessive spastic paraplegia and ataxia.

Comments to the authors

1) I find the general term “movement disorders” questionable in this study. The authors selected three major categories, which are dystonia, cerebellar ataxia and spastic paraplegia. While the inclusion of the latter is justified by the relevant overlap with cerebellar ataxia phenotype, spastic paraplegia cannot be strictly considered a “movement disorder”. On the other hand, other movement disorders were excluded from the HPO term selection, namely chorea, myoclonus and parkinsonism, just to name the most relevant ones from a genetic point of view. This should be clearly acknowledged and discussed / justified, as it should be made clear that the reported diagnostic yield cannot be generalized to rare movement disorders but only to a subset of them. I would suggest the title should also be changed to reflect this.

Response: We thank the reviewer for this remark and propose to change the title of our manuscript to “*Loss-of-function variants in the CAPN1 activator CD99L2 cause X-linked spastic ataxia,*” as this more precisely reflects the central clinical and functional findings of the present study. Furthermore, we made changes across the text to more precisely reference the disease entities under investigation and to avoid premature conclusions regarding possible additional diagnostic benefits of genome sequencing in clinical conditions not covered by this study.

2) The authors mention that panel analysis, likely performed in the early days before the widespread diffusion of WES, were not counted. However, it is likely that a subset of patients underwent NGS-panel analysis from this cohort. Can the authors provide an estimate of these figures? Did all patients who previously tested negative at panel analysis undergo WES or WGS? Conversely, were patients diagnosed through panel-based analysis included or excluded by the present study? As this is a large nationwide-based project with likely “epidemiological” implications, it is important to have a precise knowledge of inclusion and exclusion criteria.

Response: We thank the reviewer for these important comments and agree on a putative ascertainment bias introduced by previous targeted single gene or panel testing. In general, exome sequencing (ES) or genome sequencing (GS) was performed only in affected individuals without a likely diagnosis. We reviewed the available documentation indicating a

previous panel test in 262 out of 2,041 (12.8%) patients investigated by ES between 10/2016-12/2020, and in 59 out of 486 (12.1%) patients prior to GS. However, due to the sometimes long diagnostic odyssey and diversity of health care / genetic testing providers, we assume that this is a likely underestimate, especially in the early days of ES & GS. We included this information and caveat of our study in the manuscript.

3) Supplementary Figure 2: does 2B panel refer to the age at onset? Please check the letters in the panels, as they do not always correspond to the legend.

Response: We have corrected the legend, and thank the reviewer for the thorough proofreading. Age always refers to age at diagnostic testing. We have added this information.

4) Supplementary Figure 3: does 3A panel refer to the age at onset? 3B: why patients with no significant findings were tested later? Is there any correlation with the phenotype (HPO categories), or the age at onset? Also in panel 3C, explain whether this is age at onset or age at diagnosis. Same comments for figure 4.

Response: Age always refers to age at diagnostic testing. We assume, that patients with a later onset are tested later in life and in these cases, a genetic diagnosis is made less often. This is also shown in Suppl. Fig. 2b by a declining diagnostic yield with increasing age. The outcome in the different subgroups is shown in Suppl. Fig. 2c with the highest diagnostic yield for spasticity (28.1%) and the lowest for dystonia (12.1%). The outcome by age in the different subgroups is shown in Suppl. Fig. 3c. Corresponding data for GS are shown in Suppl. Fig. 4d. Here the overlapping movement disorders show the highest diagnostic yield.

5) In Supplementary Figures 2 and 4, please explain the criteria to assign “candidate genes” (yellow colour). How many of these candidate genes were validated / confirmed? They should be reported in a table.

Response: The designation ‘candidate gene’ refers to changes in genes that lack substantial publicly available evidence of a confident gene-phenotype association (e.g. report of several disease alleles in more than three unrelated families and supportive functional studies). Criteria of an assignment as ‘candidate gene’ are weighted at an individual’s level and include but are not limited to i) conservation of the gene in control cohorts (e.g. loss-of-function *de novo* change in a gene with expected loss-of-function intolerance in gnomAD: o/e score = 0), ii) bi-allelic/hemizygous predicted loss-of-function variant in linkage region / ROH in several family members or families without such variants in in-house or public controls, iii) conserved alterations in a factor with a putative mechanistic implication based on known protein function and/or specific phenotype (e.g. variant in gene coding for mitochondrial respiratory chain complex I (RCCI) subunit in a patient with mitochondrial encephalopathy and biochemically confirmed RCC1 deficiency), iv) clear variant but not matching the currently associated phenotype (alternative pathomechanism, e.g. truncated protein vs nonse-mediated decay, and v) calls for patients from collaborators. Although dependent on the specific context of the family, such changes are disclosed to the referring clinicians in a scientific report accompanying the

diagnostic report, elaborating on the available literature/evidence and perspective functional studies.

Of the 26 cases with a candidate gene in ES samples, as of today, in 21 cases the candidate genes could be subsequently established as likely causal.

Candidate gene	Reporting publication	Number of cases
C12orf66	PMID: 39824192	2 cases
HPDL	PMID: 32707086	4 cases
CD99L2	this paper	7 cases
AOPEP	PMID: 35587627	2 cases
NUDT2	PMID: 38141063	2 cases
VWA1	PMID: 33459760	1 case
PRDX3	PMID: 31782998	1 case
SPATA5L1	PMID: 34626583	1 case
KLC4	PMID: 26423925	1 case

Of the 4 cases with a candidate gene in GS samples, *CD99L2* could be further acknowledged as causal through the functional studies done for this paper.

6) Page 7, Gene-burden analysis refers to an extended prioritization list presented in Supplementary Table 3, but such Table is missing. Similarly, I cannot find Supplementary Table 4, quoted on page 7.

Response: We thank the reviewer for this remark and have attached the tables.

7) Page 7: Please check the number of index cases as they seem to be 21 (8+8+1+4), while later and in the figures the total number of families are 20.

Response: We thank the reviewer for the thorough proofreading. The index case counted here, had died at 9 months of age of probable unrelated disease and was not able for follow-up. For the clarity of presentation, we have excluded this case and corrected the numbers accordingly.

8) Legend to Figure 2: please correct “blue arrowheads in (r) and (d)” to “blue arrowheads in (r) and (s)”

Response: We have corrected this mistake, and thank the reviewer for the thorough proofreading.

9) Non-truncating variants are of interest and should be characterized more in detail. Figure 2m and 2n show that RNA levels from blood or fibroblasts of patients carrying splicing variants or a multiexon deletion are significantly reduced compared to controls (although not abolished as shown for truncating variants). Western blotting (shown for three variants) lack the wild type band but show weaker, smaller bands. Only some variants were assessed at the transcriptional level (Supplementary Figure 8). Were the bands observed on Western blotting correspond to

the transcripts detected on RNAseq? Supplementary Figure 8 should be expanded to comprehensively describe the impact of non-truncating variants at the cDNA and protein level (by the way, on figure 2, panel m refers to c.535G>C while the legend refers to c.535G>A – please check)

Response: We thank the reviewer for the additional comments and the requests to clarify the link between the transcripts detected in RNA sequencing and the protein products analysed via western blotting. In response, we have expanded and revised the representation of all viable transcripts resulting from splice-site alterations of exon deletions. This updated overview is now incorporated as an extension of Fig. 2l, and Supplementary Fig. 8 has been removed accordingly. All relevant transcripts are shown, while transcript resulting from PTVs were substantially reduced and thus assumed to undergo nonsense-mediated decay.

By comparing the immunogenic regions of the N-terminal and C-terminal CD99L2 antibodies used in our western blot analyses, we can reasonably infer – though not conclusively prove – that the transcripts lacking exons 10 and 11, such as those arising from the c.535G>C (p.Gly179Arg) variant (see transcript a)) and c.655+1G>A (see transcript a) and c)), may be the templates for the CD99L2 protein forms detected in Fig. 2r and 2s.

Furthermore, the detection of a protein band in fibroblasts from a patient carrying the c.497-612_655+2267delinsC variant using both N-terminal and C-terminal antibodies supports the interpretation that the template transcript encodes an internally truncated CD99L2 protein lacking exons 8 and 9. This is consistent with the revised schematic in Fig. 2l (transcript a) for variant c.497-612_655+2267delinsC), the protein signal in Fig.2t, and Supplementary Fig. 7o.

In addition to modifying the figure and legend accordingly, we have revised the manuscript text to include a new paragraph titled “Consequences of identified CD99L2 variants on transcript and protein levels” in the Result section, which now reads as follows:

To further support the functional relevance of identified variants, we conducted transcriptome sequencing (TS) and western blot studies on available patients’ biospecimens. These included analyses of PAXgene Blood RNA Tubes, representing all identified variants, and fibroblasts from affected individuals carrying six different variants (Fig. 2o-t; Supplementary Fig. 7j-o). For all PTVs studied, CD99L2 mRNA levels were significantly reduced compared to controls, in line with nonsense-mediated decay (NMD) of mutant transcripts. When quantified as read counts, CD99L2 mRNA levels of individuals with predicted splice site alterations and exon deletions were not significantly altered. However, no wild-type transcripts were observed, only transcripts indicating exon skipping, intron retention, and/or variable use of alternative splice sites, as well as a potential upregulation of the shorter transcript ENST00000418547.2 (Fig. 2l; transcript c) of variant c.655+1G>A). Apart from the exon 8/9 deletion, which results in an in-frame transcript missing the entire transmembrane domain, all other aberrant mRNAs predict mainly prematurely truncated proteins. The use of alternative splice sites as shown in Fig. 2l; transcript b) of variant c.535G>C and c.655+1G>A was only seen in a smaller subset of reads.

To analyse the consequences of the identified CD99L2 variants at the protein level, we performed western blotting of fibroblast-derived protein extracts using antibodies directed against either the N-terminus (targeting residues 27-117) or the C-terminus (targeting residues 209-254) of the protein (Fig. 2o-t; Supplementary Fig. 7j-o). In controls, a strong band was detected between 37 kDa and 50 kDa using the N-terminal antibody – higher than the predicted molecular weight of 28 kDa for full-length CD99L2.

However, the specificity of this signal was confirmed by parallel detection using the C-terminal antibody.

In patients carrying PTVs, CD99L2-specific bands were entirely absent (Fig. 2o-q; Supplementary Fig. 7j-l). In contrast, patients with splice-site alterations or a deletion of exons 8/9 showed faint bands of lower molecular weight when probed the N-terminal antibody (Fig. 2r-t) and – exclusively in the case of the exon 8/9 deletion – also with the C-terminal antibody (Suppl. Fig. 7m-o). These may represent internally or C-terminally truncated CD99L2 protein forms, likely corresponding to translation products from the aberrant transcripts detected in the transcriptome analysis (Fig. 2l).

In conclusion, all CD99L2 variants investigated led to a near-complete loss of full-length CD99L2, supporting a loss-of-function as the likely pathomechanism.”

The error in legend to Fig. 2 has been corrected to c.535G>C.

10) Please comment why autophagy inhibition results in a reduction of ubiquitinated CD99L2 while the unmodified protein increases (Figure 3d).

Response: We thank the reviewer for this request and appreciate their careful observation. Our data, derived from analyses of both full-length and truncated CD99L2 constructs, clearly demonstrate that CD99L2 undergoes bimodal degradation via both the proteasomal and autophagosomal pathways. The C-terminus appears to mediate proteasomal degradation, while the N-terminus is associated with autophagosomal degradation, although the proteasomal route remains the predominant pathway. As demonstrated in many studies, including ours, there is a strong interplay between autophagy and the proteasomal system (10.1016/j.tplants.2023.01.013). Inhibition of one pathway often triggers compensatory activation of the other. This is reflected in our data, where we observe decreased K48-linked polyubiquitin levels upon BafA1 treatment and increased LC3B-II levels following MG132 administration (see Fig. 3b; Reviewer Fig. 1). We believe the modified (i.e., ubiquitinated) forms of CD99L2 shown in Fig. 3d are primarily targeted by the UPS. Accordingly, these forms accumulate strongly upon proteasome inhibition but are reduced upon autophagy inhibition - likely due to enhanced UPS-mediated turnover as a compensatory response. Conversely, the full-length, unmodified CD99L2 accumulates under autophagy inhibition, suggesting that this fraction may be preferentially degraded via autophagy, potentially through interactions specific to membrane proteins or organelles, rather than via a ubiquitination-dependent mechanism. While this explanation is plausible, it requires further investigation, which we hope to address in future studies focused on the precise modulatory roles of post-translational modifications and catabolic pathways in regulating CD99L2 turnover.

To additionally address this observation, we have included the following sentences to the respective result section:

“Notably, proteasomal inhibition by MG132 was the most effective in elevating CD99L2 levels, leading to an approximately twelve-fold increase of fl-CD99L2 compared to a five-fold increase upon autophagy inhibition using bafilomycin A1 (BafA1) (Fig. 3d). Moreover, while BafA1 treatment increased levels of the unmodified full-length CD99L2, the levels of the modified, potentially ubiquitinated form declined, likely reflecting of a compensatory upregulation of proteasomal activity in response to autophagy inhibition.”

Reviewer Figure 1: Inhibition of the UPS or autophagy triggers reciprocal compensatory activation of the other pathway. Western blot analysis of K48-linked polyubiquitin (K48-polyUb) and LC3B-I/II in 293T cells transfected with myc-His₆-CD99L2, following 24 h treatment with MG132 or bafilomycin A1 (BafA1). β-actin was used as a loading control. Densitometric quantification of K48-polyUb and LC3B-II levels normalized to β-actin. *n* = 6 biologically independent experiments. Statistical analysis by one-sample *t*-test or Student's *t*-test, as appropriate. Bars represent mean ± s.e.m. **P* ≤ 0.05; ***P* ≤ 0.01; ****P* ≤ 0.001; *****P* ≤ 0.0001.

11) Legend to figure 3: please move description of grey arrowheads/white arrowheads/black arrowheads to panels a-c.

Response: We have added a description of the grey, white, and black arrowheads after the text describing panel a.

12) Figure 4h-j: can the authors explain this divergent behavior of truncated CD99L2 in response to proteasome or autophagy inhibition?

Response: We thank the reviewer for this insightful question. We believe that the N- and C-terminal domains, as well as the subcellular localization of the protein or its domains, play a crucial role in mediating CD99L2 degradation, which we have characterized as bimodal in our manuscript. This is likely facilitated by the presence of specific sites for post-translational modifications – such as distinct lysine residues for ubiquitination – and the presence or absence of defined recognition motifs for the two degradation pathways. Further research on CD99L2 is needed to identify which specific sites and motifs are key determinants of its proteostatic fate.

To address this point more clearly in the manuscript, we have expanded the description of this experiment, which now reads as follows:

“Given the previously observed bimodal degradation of full-length CD99L2, we next explored whether specific domains influence its proteostatic fate. Notably, CD99L2₁₋₁₈₅ accumulated upon autophagy inhibition, while CD99L2₂₀₇₋₂₆₂ levels increased following proteasome inhibition (Fig. 6h-j; Supplementary Fig. 12f-i). These findings suggest that CD99L2's bimodal degradation is domain-dependent, with the N-terminal region driving its autophagosomal clearance and the C-terminal domain directing proteasomal degradation.

13) Figure 4k: it would be useful to have a brief explanation of this schematic cartoon to help the readers draw the right conclusion about the postulated mechanism.

Response: We are grateful to the reviewer for pointing this out. We have amended the legend to Fig. 4k (now Fig. 6k) by including a brief description of the schematic, which outlines the postulated mechanism. The respective section reads as follows:

“Model depicting the functional interaction between CD99L2 and CAPN1, and the proteostatic regulation of CD99L2. Inactive CAPN1 binds to unmodified full-length CD99L2 and becomes activated. Active CAPN1 then cleaves CD99L2, removing its C-terminus, which in turn inhibits the further activation of additional CAPN1 molecules. Furthermore, an interplay between E3 ubiquitin ligases (E3) and deubiquitinases (DUB) maintains a balance between ubiquitinated and unmodified full-length CD99L2. This ensures continuous degradation of the protein via the ubiquitin proteasome system (UPS), or to a lesser extent, autophagy, thereby preventing excessive activation of the calpain system. CP, cytoplasm; EC, extracellular space.”

14) Supplementary Figure 9: it seems that anti-CD99L2 antibody does not recognize any endogenous protein either in 293T cells or SH-SY5Y cells (mock transfected lane is perfectly clear not only in the myc-antibody panels but also in the CD99L2 panels and overexposure shows a very faint modified band). Is the protein expected to have a very rapid turnover in cells? Figure 3C seems to show that even inhibition of proteasome or autophagy do not result in increased levels of endogenous protein in mock-transfected cells. However, in Supplementary Figure 12 it seems that endogenous CD99L2 is detected in control and SPG76 fibroblasts. Please provide an explanation for these observations or clarify in the text.

Response: We thank the reviewer for this insightful comment. Indeed, the difference in expression levels between endogenous and overexpressed CD99L2 is substantial, making it extremely challenging – if not impossible – to track the fate of the endogenous protein when samples are loaded side by side on the same Western blot. The strong signal from the overexpressed CD99L2 even caused bleed-over into adjacent lanes. However, only with such high expression levels were we able to clearly visualize the unmodified full-length form of CD99L2, as well as its potential fragments.

Nevertheless, our data in Fig. 4a and 4b, as well as Supplementary Fig. 8c and 8d, demonstrate that CD99L2 is detectable in its ubiquitinated/modified form in both 293T and SH-SY5Y cells. Moreover, we show that this modified form can be modulated, for example, via CAPN1 knockdown (Fig. 4a). As the reviewer suggests, low baseline expression, rapid turnover, and cell- or tissue-specific differences in CD99L2 expression and degradation may all contribute to the challenges with analyzing this protein. We propose that CD99L2’s high turnover rate may serve as an important regulatory mechanism, potentially preventing excessive calpain activation, as hypothesized in our schematic model. To address this further, we repeated the analysis in mock-transfected cells treated with MG132 and BafA1. These experiments confirm that CD99L2 is detectable in its modified form and that this form is indeed modulated by treatment, specifically upon proteasomal inhibition (see Reviewer Fig. 2).

In line with this, we conducted a pilot experiment to assess the half-life of CD99L2 in 293T cells using a cycloheximide (CHX) chase assay in the presence or absence of the deubiquitinase ataxin-3, which plays a role in the UPS. While replication and validation of this

assay is warranted and will be included in our follow-up studies, our initial data indicate that CD99L2 has a mean half-life of approximately 18 hours in 293T WT cells, which is substantially shorter than the 30-hour half-life predicted by the ExPASy ProtParam tool (web.expasy.org/protparam) (see Reviewer Fig. 3).

Finally, we have revised the corresponding section in the Results to include a more detailed explanation of our findings, which now reads as follows:

“To investigate the discrepancy between the calculated (~28 kDa) and observed (37–50 kDa) molecular weight of CD99L2 in control fibroblasts (Fig. 2o-t; Supplementary Fig. 7j-o), we first replicated these findings in 293T cells and SH-SY5Y neuroblastoma cells, both at baseline and upon overexpression of a CD99L2 construct. Only upon overexpression, at levels several-fold higher than the endogenous levels, we were able to detect the apparently unmodified full-length form of CD99L2, along with potential fragments, while the modified form represented the predominant species (Fig. 3a; Supplementary Fig. 8a-d).”

Reviewer Figure 2: Inhibition of the UPS increased levels of ubiquitinated CD99L2 in 293T cells. **a**, Western blot analysis of K48-linked polyubiquitin (K48-polyUb) and LC3B-I/II in 293T cells transfected with myc-His₆-CD99L2, following 24 h treatment with MG132 or bafilomycin A1 (BafA1), shows efficient inhibition of both pathways. GAPDH served as the loading control. **b**, Western blot and densitometric analysis of ubiquitinated CD99L2 in the same samples. GAPDH served as the loading control. Grey arrowheads indicate ubiquitinated full-length (Ub) CD99L2. Ub-CD99L2 levels were normalized to GAPDH. *n* = 4 biologically independent experiments. Statistical analysis by one-sample *t*-test or Student’s *t*-test, as appropriate. Bars represent mean ± s.e.m. **P* ≤ 0.05; ***P* ≤ 0.01.

Reviewer Figure 3: Assessment of CD99L2 half-life in 293T cells using a cycloheximide chase assay. **a**, Western blot analysis of K48-linked polyubiquitin (K48-pUb) and ubiquitinated CD99L2 (Ub-CD99L2) in 293T cells subjected to an 8 h cycloheximide (CHX) chase assay. GAPDH and SYPRO Ruby total protein stain served as loading controls. Grey arrowheads indicate ubiquitinated (Ub) CD99L2, while the white arrowhead marks unmodified full-length (fl) CD99L2. **b**, Densitometric analysis of K48-pUb and Ub-CD99L2. Ub-CD99L2 levels were normalized to SYPRO Ruby signal and then to the values at the 0 h time point. $n = 3$ biologically independent experiments. A one-phase decay model with robust regression fitting was applied for the nonlinear regression analysis.

15) The authors discuss a cellular mislocalization of truncated forms of CD99L2 compared to wild type. By assessing colocalization with an ER-marker, it seems that truncated forms of the protein have increased co-localization with ER compared to wild type, however this is not explained in the text. Why the authors specifically assessed colocalization with ER? Have they also tested colocalization with CAPN1?

Response: We thank the reviewer for this request and their insightful comments. The initial observation of mislocalization was made without the use of additional counterstaining or cell compartment-specific reporters. However, we observed a marked contrast between the expected plasma membrane localization of full-length CD99L2 and the granular, cytoplasmic (non-nuclear) distribution of the N-terminal CD99L2₁₋₁₈₅ fragment, while the smaller C-terminal CD99L2₂₀₇₋₂₆₂ fragment displayed a diffuse distribution in both the cytoplasm and nucleus. Given that the pattern of CD99L2₁₋₁₈₅ resembled the endoplasmic reticulum (ER), we performed co-transfection with a dsRed-tagged ER reporter. This confirmed colocalization, indicating that the N-terminal fragment is mislocalized to the ER (now Supplementary Fig. 12b). To better illustrate this observational sequence, we have now included a panel with the first microscopy-based analysis of the CD99L2 truncation constructs, performed without co-expression of the ER reporter (new Supplementary Fig. 12a) and updated the referencing to the figure accordingly.

The observed mislocalization of CD99L2₁₋₁₈₅ may result from the retention of the plasma membrane-targeting signal peptide at the N-terminus (residues 1 – 25), coupled with the absence of the transmembrane domain in this fragment. Since we had not previously mentioned the N-terminal signal peptide in our manuscript or schematics, we have now revised the manuscript accordingly to include this detail.

We had not initially tested co-localization of CD99L2 and CAPN1 via microscopy. However, following the reviewer's suggestion, we have now performed the relevant co-immunostainings. These new results show that all CD99L2 constructs partially share the subcellular space with CAPN1, offering potential contact sites for an interaction. Nevertheless, as demonstrated by our co-immunoprecipitation experiments (Fig. 6f, g), this partial spatial overlap is insufficient to induce calpain activation or binding. The new co-staining results have been included in our manuscript as the new Fig. 6b, and we have updated the result section accordingly.

In addition, we have expanded the corresponding section in the results to provide a more detailed explanation, as suggested by the reviewer. The revised section now reads as follows:

“Our analysis underscored the importance of CD99L2's structural integrity: while all constructs showed limited colocalization with CAPN1, CD99L2₁₋₁₈₅ displayed a compartmentalized cytoplasmic pattern reminiscent of the endoplasmic reticulum (ER), which was confirmed via co-expression with a DsRed-tagged ER reporter. In contrast, CD99L2₂₀₇₋₂₆₂ exhibited a more diffuse distribution in the cytoplasm and, to a lower extent, the nucleus. Full-length CD99L2, as expected, localized to the plasma membrane (Fig. 6b; Supplementary Fig. 12a, b).”

Reviewer #2 (Remarks to the Author):

This manuscript describes the approach to increase diagnostic power for rare diseases by rationally combining the measures to detect potential cause at the level of gene sequence and their association with clinical phenotypes. In their experimental settings using movement disorders as study subjects, improvements in diagnostic yields were suggested and the novel disease gene, CDL99L2, was identified. Then, properties of the gene product were explored demonstrating that CD99L2 is constitutively ubiquitinated and its turnover is regulated by proteasome and autophagy. In addition, CD99L2 interacts with CAPN1, a catalytic subunit of intracellular protease calpain-1, and activates it. Further, truncated mutants of CD99L2, either the N- or C-terminal, were shown to be defective in localization, calpain interaction/activation as well as ubiquitination. It is suggested that pathogenicity of the CDL99L2 variants identified in the study is manifested via dysregulation of calpain and/or compromise in degradative regulation of CDL99L2.

The first part of the study shows the impact of systematically performed evaluation of the huge sets of sequence data. Identification of a novel causative gene, CDL99L2, for the movement disorder is a great outcome without question. In the following experiment, basic properties of CDL99L2 was examined, however, the results as shown in Fig. 4 are not sufficient to support what the authors claim in discussion, i.e., functional investigation on CDL99L2 is missing. Therefore, it is not clear if there is any real association between the proteolytic systems, either proteasome, autophagy or calpain, and CD99L2. To demonstrate the relevance of the present result in terms of pathogenic mechanism stemming from the CDL99L2 variants, following

points, at minimum, should be clarified; is there any functional relationship between ubiquitination and CAPN1-interaction of CD99L2?, in terms of calpain-1 activation function of CD99L2, what cellular functions are compromised in the cells from patients harboring CD99L2 mutations? (is activation of calpain-1 impaired?), what is functional impact of ubiquitination of CD99L2 function?

Response: We would like to express our sincere gratitude to the reviewer for their thoughtful assessment of our manuscript, as well as for the valuable comments and recommendations aimed at improving the significance and conclusiveness of our study, both in major and minor aspects. While we firmly believe that our molecular analysis, which validated and elucidated key aspects of CD99L2 function and the CD99L2–calpain axis – including structural features, post-translational modifications, protein interactions, degradation pathways, and downstream proteolytic activation of the calpain system – convincingly demonstrates the relevance of the proposed pathway, we acknowledge that there is still room for refinement and improvement. Accordingly, we have made every effort to thoroughly address all points raised by the reviewer to the best of our knowledge and within the limits of our currently available resources.

We have addressed the major points raised by the reviewer as follows:

1. Is there any functional relationship between ubiquitination and CAPN1-interaction of CD99L2?

Response: The reviewer raised an insightful and valid question regarding the relationship between the ubiquitinated form of CD99L2 and its interaction with CAPN1, which we had previously addressed only hypothetically in our schematic summary (now Figure 6k). To explore this experimentally, we utilized our lysine-free CD99L2 construct (CD99L2_{K0}) and conducted new co-immunoprecipitation (co-IP) experiments and fluorescence microscopy analysis to assess the effect of lysine depletion – and thus loss of ubiquitination – on the interaction between CD99L2 and CAPN1.

Interestingly, our western blot analysis revealed that the absence of lysine residues (via lysine-to-arginine mutagenesis), and therefore the lack of CD99L2 ubiquitination, resulted in a markedly increased binding to CAPN1. In contrast, the negligible interaction with CAPN2 remained unchanged (see new Fig. 5b).

To ensure that this increased binding was not due to altered subcellular localization, we examined the localization of CD99L2_{K0} via fluorescence microscopy. The lysine-free construct maintained the same plasma membrane-specific localization as the wild-type protein (see new Fig. 5a), indicating that ubiquitination does not influence CD99L2 targeting. Consequently, the limited, membrane-proximal interaction site between CD99L2 and CAPN1 was not altered, as demonstrated by our co-staining with CAPN1. Therefore, the observed increase in CAPN1 binding is not attributable to changes in protein localization.

These new findings are presented in the new Fig. 5 and in the updated Results section of the manuscript, which reads as follows:

“After demonstrating that CD99L2 directly binds to CAPN1 and induces calpain system activation, we explored whether its ubiquitination capacity contributes to this function.

To investigate this, we overexpressed both myc-tagged wild-type CD99L2 and its lysine-free variant CD99L2_{K0} in 293T cells. Fluorescence microscopy revealed no differences in subcellular localization of the two CD99L2 variants, including their limited colocalization with CAPN1 at membrane-distal contact sites, indicating that lysine residues – and their ubiquitination – do not alter CD99L2 targeting to the plasma membrane (Fig. 5a).

To assess changes in protein interaction, we performed co-IP via the myc tag of the CD99L2 constructs. Western blot analysis revealed a markedly increased binding affinity of CD99L2_{K0} to CAPN1 compared to wild-type CD99L2 (Fig. 5b). Notably, the absence of lysine residues in CD99L2 did not enhance its weak binding to CAPN2.”

2. In terms of calpain-1 activation function of CD99L2, what cellular functions are compromised in the cells from patients harboring CD99L2 mutations?

Response: Although our functional analysis using CD99L2 truncation constructs provides some indications of the molecular consequences expected in the patient context, we agree with the reviewer that downstream pathway analyses in patient-derived models would offer more substantial insight into the cellular functions compromised by CD99L2 variants and the mechanisms underlying the observed phenotype. While a neuronal system would be optimal to assess these potential dysregulations in a more physiological context—a consideration for future studies—we have now performed RNA sequencing–based transcriptome analysis of primary fibroblasts derived from CD99L2 patients and healthy controls, followed by pathway and enrichment analyses.

We observed significant dysregulation of more than 500 genes in patient fibroblasts. Notably, biological process and cellular compartment enrichment analyses revealed a characteristic dysregulation of genes involved in neuronal and synaptic function. Among the top 10 enriched biological processes and cellular compartments, the majority were related to neuronal and synaptic functions. These findings are consistent with previous studies investigating the effects of *CAPN1* loss (Wang et al., 2016; PMID: 27320912) and with recent work on the neuronal role of CD99L2 (PMID: 39808524), in which synaptic function was primarily affected and linked to cerebellar ataxia or hippocampal function.

Together, these results strongly support our hypothesis that CD99L2 influences neuronal function and may play a role in the CAPN1-related regulatory axis.

We have incorporated these data as new Fig. 7 and Supplementary Fig. 13, including their respective legends, along with the following textual additions in the manuscript:

Results:

“CD99L2 patient fibroblasts show transcriptional dysregulation of genes associated with neuronal function

After functionally dissecting the CD99L2 protein, we next analysed the downstream consequences of CD99L2 variants to better explain the neurological phenotype observed in affected patients. To this end, we performed transcriptomic profiling of fibroblasts from patients carrying PTVs, splice-site variants, or deletions, compared to fibroblasts from unaffected matched controls. RNA sequencing identified 515 significant differentially expressed genes (DEGs), including 191 downregulated and 324 upregulated transcripts (Fig. 7a). Since the physical interaction of CD99L2 and the calpain system occurs at the protein level, we specifically assessed potential

transcriptional dysregulations of related genes. However, no significant expression changes were detected for *CAPN1*, *CAPN2*, *CAPNS1* (encoding CSS1), *CAST*, or *SPTAN1* (encoding the calpain substrate α -spectrin) (Supplementary Fig. 13a).

To identify functional themes among the DEGs, we performed STRING-based gene ontology (GO) enrichment analysis (Szklarczyk et al., 2023). Among the top 10 enriched biological processes, 8 were directly linked to neuronal function. Similarly, 6 of the top 10 enriched cellular compartments were related neurons, particularly the synapse (Fig. 7b, c). Beyond the general GO term “cell junction”, the strongest enrichment was observed for “synapse,” which encompassed 55 DEGs showing marked dysregulation and a high degree of functional interaction (Fig. 7d, e; Supplementary Fig. 13d).

Although fibroblasts are not the model of choice for directly studying neuron-specific pathology, the consistent dysregulation of neuron- and synapse-related genes highlights the likely impact of CD99L2 variants on neuronal function and degeneration. In light of previous findings that loss of *CAPN1* impairs synaptic function and triggers cerebellar ataxia (Wang et al., 2016), and given that CD99L2 may regulate synapse development (Kang et al., 2025), our data suggests that perturbed CD99L2 function could exert similarly detrimental effects in the patient brain.”

Discussion:

“Our transcriptome analysis of patient-derived fibroblasts revealed a pronounced and specific dysregulation of genes, with strong enrichment for biological processes linked to neuronal function, particularly the synapse. This transcriptional signature overlaps with synaptic pathways known to be affected by the loss of *CAPN1*, whose LOF variants cause SPG76 and forms of cerebellar ataxia (Gan-Or et al., 2016; Méreaux et al., 2021; Wang et al., 2016).”

3. Is activation of calpain-1 impaired [in the cells from patients harboring CD99L2 mutations]?

Response: We thank the reviewer for this relevant question. As CD99L2-mediated CAPN1 activation appears to be a conserved mechanism across our cell model systems, the absence of the protein is expected to reduce calpain activation to a certain extent. To address this, we followed the reviewer’s suggestion and investigated calpain-mediated α -spectrin fragmentation, as well as CAPN1 and CAST levels, in the fibroblast lines available in our laboratory. Western blot analysis did not reveal significant changes in CAPN1 or CAST levels, although some variability was observed. However, cleavage of α -spectrin was significantly reduced in patient-derived fibroblasts. This finding is consistent with the expected effect of CD99L2 absence or loss, when considered as a calpain activator.

We have included this analysis in the new Supplementary Fig. 13b, c and added a corresponding description of our findings in the Results section, which now reads as follows:

Furthermore, we analysed protein levels of CAPN1 and CAST, as well as calpain-mediated cleavage of α -spectrin, in fibroblasts derived from CD99L2 patients and healthy controls. While no significant alterations in CAPN1 or CAST levels were detected, spectrin cleavage

was reduced in patient fibroblasts, consistent with the absence of the CAPN1 activator CD99L2 (Supplementary Fig. 13b, c).

4. What is functional impact of ubiquitination of CD99L2 function?

Response: Following our results indicating the increased CAPN1 binding of CD99L2_{K0}, we subsequently investigated whether this enhanced interaction leads to stronger activation of the calpain system, in order to highlight the functional relevance of CD99L2 ubiquitination, as requested by the reviewer. By analysing the protein levels and calpain-mediated proteolysis of CAPN1, CAST, α -spectrin, and CSS1, we were able to demonstrate a clear increase in calpain activation (see new Fig. 5c, d). These findings suggest that the ubiquitination may not only regulate CD99L2 degradation via the proteasomal pathway, but also modulate its function as a calpain activator.

We have included this data in the new Fig. 5 and in the respective paragraphs of the Results and Discussion section, which read as follows:

Results:

“Next, we assessed the impact of lysine residue loss on CD99L2’s ability to activate calpains. Western blot analysis of calpain system markers – including CAPN1 and CAST levels, as well as α -spectrin and CSS1 cleavage – indicated a further amplification of proteolytic activation in cells expressing CD99L2_{K0} (Fig. 5c, d).

Together, these findings suggest that the ubiquitination of CD99L2 modulates its interaction with CAPN1 and dampens calpain activation, potentially serving as a regulatory mechanism to prevent excessive and harmful calpain activity within cells.”

Discussion:

“Specifically, ubiquitination of CD99L2 appears to reduce its binding to CAPN1 and attenuate calpain system activation, consistent with established roles of this PTM in modulating protein-protein interactions (Tracz & Bialek, 2021). This regulatory mechanism modulation may help prevent excessive proteolytic activity, which has been implicated in neurodegenerative diseases such as polyglutamine disorders.”

We have addressed the reviewer’s comments on minor points as follows:

1. (line 176) The case number for FA should be 1,067 (as in Supl. Fig. 1).

Response: We thank the reviewer for this remark. We have corrected the number respectively.

2. (line 270) The change in molecular weight by ubiquitination is ca. 22 kDa at maximum corresponding to 3 ubiquitins, therefore, “highly” is unnecessary.

Response: We agree with the reviewer’s assessment and apologize for the inadequate wording regarding the ubiquitination of CD99L2. Our intention was to emphasize that the CD99L2 protein appears in its ubiquitinated form, rather than in its unmodified state. The

section containing the term “highly” has been removed, and we have avoided using it elsewhere.

3. (line 226 in supplement) What mutation(s) does “lysine-free” stand for?

Response: We thank the reviewer for the request and clarified the characteristic of our construct respectively. The reader will find the following description in the main text:

“We validated these findings by mutating all 14 lysine residues in CD99L2 (Fig. 3g) to arginine (CD99L2_{K0}) to prevent canonical ubiquitination, which successfully resulted in a shift from the higher molecular weight form of CD99L2 to its unmodified full-length form (Fig. 4h, i).”

We have also amended the respective figure legend:

“**h**, Western blot analysis of 293T cells overexpressing wild-type (WT) myc-His₆-CD99L2 and a lysine-free (K0) variant in which all 14 lysine residues were mutated to arginine, using a myc tag-specific antibody. GAPDH was used as a loading control.”

4. (line 445 in supplement) (l) should be (i).

Response: Thank you. We have corrected this mistake in the figure legend to Supplementary Fig. 9 (now Supplementary Fig. 8).

5. In Figure legends, definitions for abbreviation and labelling (e.g., SBDP, “full”, “*” for statistical significance, etc.) should be provided when they first appeared.

Response: We thank the reviewer for noticing these inconsistencies. We have edited our figure legends and included the missing explanation of abbreviations and labelling according to their first appearance in the respective figure.

6. Methods used for statistical significance should be consistently provided (missing in some figure legends).

Response: We have checked and updated the figure legends in this regard and included the missing information where necessary.

7. What does labelling “*” in pedigree schemes in Fig. 1(a-k) and Fig. S7(a-i) indicate?

Response: We thank the reviewer for pointing out this missing information. In both figure legend, we added the following explanatory sentence:

“Asterisks (*) indicate fully characterised patients.”

8. What is the rationale in testing spectrin cleavage and including CAPN10 in the experiment shown in Fig. 3 etc.?

Response: In the previous version of the manuscript, the conciseness of our descriptions may have left some of the rationale behind our analysis insufficiently clear. We have addressed this in the revised manuscript by providing more detailed explanations throughout. Specifically, regarding the reviewer's request for clarification of the marker proteins α -spectrin and CAPN10, we have integrated the following sections into the corresponding Result paragraph:

Regarding spectrin cleavage:

"In our co-IP experiments, we observed increased autoproteolysis of CAPN1, reduced levels of the endogenous proteinaceous inhibitor calpastatin (CAST), and a fragmentation-related decrease in full-length CSS1, suggesting activation of the calpain system upon CD99L2 transfection. To confirm this, we repeated CD99L2 overexpression in 293T and SH-SY5Y cell lines and quantitatively analysed key protein markers, including the known calpain substrate α -spectrin, whose calpain-mediated proteolysis generates a characteristic ~150 kDa fragment. All analysed proteins exhibited enhanced proteolytic processing in both cell models (Fig. 3l, m; Supplementary Fig. 9e-h), corroborating CD99L2-induced activation of calpains."

Regarding CAPN10:

"Previous affinity capture-based mass spectrometry identified CD99L2 as an interactor of calpain-1 (CAPN1), a calcium-activated cysteine protease involved in neuronal plasticity and implicated in recessive spastic paraplegia and cerebellar ataxia (Huttlin et al., 2021; Baudry et al., 2016). To validate these findings, we performed co-immunoprecipitation (co-IP) experiments using two differently tagged CD99L2 constructs overexpressed in 293T cells. Western blot analysis confirmed that CD99L2 directly interacts with CAPN1 and apparently co-precipitates its regulatory small subunit CSS1 (Fig. 3j, k; Supplementary Fig. 9a-d). A potential weak binding to calpain-2 (CAPN2) was also observed (Extended Data Fig. 10d), whereas no interaction was detected with the non-classical calpain-10 (CAPN10), which structurally differs from CAPN1 by lacking the C-terminal calcium-binding penta-EF-hand domain—and thereby the ability to interact with CSS1 (Ono et al., 2016)—or with the endogenous proteinaceous inhibitor calpastatin (CAST)."

9. What does "mislocalization" (as in descriptions for Fig. 4b and Fig.13a) mean exactly?

Response: As mentioned above, the previous version of our manuscript was highly concise and thus did not allow for a detailed explanation of our observations. Pertaining to the findings previously shown in Fig. 4b and Fig. 13a (now revised, extended, and re-labeled as Fig. 6b and Suppl. Fig. 12a, b), we have incorporated the following explanatory passage into the corresponding Results section:

"Our analysis underscored the importance of CD99L2's structural integrity: while all constructs showed limited colocalization with CAPN1, CD99L2₁₋₁₈₅ displayed a compartmentalized cytoplasmic pattern reminiscent of the endoplasmic reticulum (ER), which was confirmed via co-expression with a DsRed-tagged ER reporter. In contrast, CD99L2₂₀₇₋₂₆₂ exhibited a more diffuse distribution in the cytoplasm and, to a lower extent, the nucleus. Full-length CD99L2, as expected, localized to the plasma membrane (Fig. 6b; Supplementary Fig. 12a, b)."

10. The title could be modified to better represent the article's content.

Response: We concur with the reviewer that modifying of the manuscript title is beneficial. Accordingly, we have changed the title to “*Loss-of-function variants in the CAPN1 activator CD99L2 cause X-linked spastic ataxia*”, as it more precisely reflects the central clinical and functional findings of the present study.

Reviewer #1 (Remarks to the Author)

I appreciate the work made by the authors to improve the manuscript and address the criticisms raised by the reviewers. A lot of additional experimental work has been added and several issues have been clarified. I am overall satisfied of the data presented here.

Response: We thank the reviewer for the positive feedback and appreciate the recognition of our efforts to strengthen the manuscript in response to the reviewers' comments.

Reviewer #2 (Remarks to the Author):

The revised manuscript shows great improvement in the overall construction as well as data presentation. It is greatly appreciated that the authors have added much work and I believe that an important lead for the future studies is established.

Response: We thank the reviewer for the positive assessment and greatly appreciate the recognition of the additional work incorporated during revision. We are pleased that the reviewer views the study as establishing an important lead for future research.

A few minor corrections:

Page 6, line 165; the number "1,076" should be "1,067" (as in line 181).

Response: We thank the reviewer for noticing this mistake. We have corrected the number accordingly.

Page 7, line 224; considering the expression "Of the 3 associations identified, 2... were already known...and 1... was ... unidentified (line 236-238), "three known" should be "two known"?

Response: We appreciate the reviewer's careful reading. We have revised the wording accordingly.

Page 9, line 331; should be "Fig. 3h, i"?

Response: We thank the reviewer for pointing this out. We had originally updated the figure reference following the reviewer's correct remark. However, because we subsequently rearranged and split figures to meet the journal's formatting requirements, the correct reference in the final layout is now again "Fig. 4h, i."

Changing the order of Supplementary Tables could better suit the flow of the manuscript; the present Table S2 showing a list of used antibodies could be Table S4 since the content is not referred to in the main text and stands as a component in supplementary methods.

Response: We thank the reviewer for this helpful suggestion. Because the former Supplementary Tables S3 and S4 were redesignated as Supplementary Data 1 and Supplementary Data 2 to comply with the journal's formatting guidelines, only two

Supplementary Tables remain. For this reason, the antibody table continues to be numbered “Supplementary Table 2.”